# Development of Inter-Grid Cell Lateral Unsaturated and Saturated Flow Model in the E3SM Land Model (v2.0)

Han Qiu[1], Gautam Bisht[1], Lingcheng Li[1], Dalei Hao[1], and Donghui Xu[1]

[1]Atmospheric Sciences and Global Change Division, Pacific Northwest National Laboratory, Richland, WA, USA

**Correspondence:** Han Qiu (han.qiu@pnnl.gov)

**Abstract.** The lateral transport of water in the subsurface is important in modulating the terrestrial water-energy distribution. Although few land surface models have recently included lateral saturated flow within and across grid cells, it is not a default configuration in the Climate Model Intercomparison Project version 6 experiments. In this work, we developed the lateral subsurface flow model within both unsaturated and saturated zones in the Energy Exascale Earth System Model (E3SM) Land Model version 2 (ELMv2.0). The new model, called $ELM_{lat}$, was benchmarked against PFLOTRAN, a 3D subsurface flow and transport model, for three idealized hillslopes that included a convergent hillslope, divergent hillslope, and titled V-shape hillslope with variably saturated initial conditions. $ELM_{lat}$ showed comparable performance againt PFLOTRAN in terms of capturing the dynamics of soil moisture and groundwater table for the three benchmark hillslope problems. Specifically, the mean absolute errors (MAE) of the soil moisture in the top ten layers between $ELM_{lat}$ and PFLOTRAN were within $1\% \pm 3\%$ and the MAE of water table depth were within $\pm 0.2$ [m]. Next, $ELM_{lat}$ was applied to the Little Washita experimental watershed to assess its prediction of groundwater table, soil moisture, and soil temperature. The spatial pattern of simulated groundwater table depth agreed well with the global groundwater table benchmark dataset generated from a global model calibrated with long term observations. The effects of lateral groundwater flow on the energy flux partitioning were more prominent in low land areas with shallower groundwater tables, where the difference of simulated annual surface soil temperature could reach $0.3 - 0.4$ [°C] between ELMv2.0 and $ELM_{lat}$. Incorporating lateral subsurface flow in ELM improves the representation of the subsurface hydrology which will provide a good basis for future large-scale applications.

## 1 Introduction

Groundwater, which stores $\sim 30\%$ of the world's freshwater, plays an important role in the global hydrologic cycle and is a critical water resource for the environment and human systems. As the bottom boundary, groundwater moderates soil moisture that is extracted by vegetation roots while is recharged by water percolation and acts as an important buffer in the water cycle (Maxwell et al., 2007; Fan et al., 2007; Miguez-Macho et al., 2007; Fan, 2015). Groundwater also interacts with rivers by supporting the base flow or receiving the percolated water from rivers and feeds the groundwater-dependent wetlands (De Graaf et al., 2014; de Graaf et al., 2017; Condon and Maxwell, 2019; Qiu et al., 2019, 2020). Groundwater is a major freshwater resource for drinking and irrigation and has been used for various industrial purposes (Döll et al., 2012; Wada et al., 2011). With the surging growth of population and water demand, overexploitation of groundwater resources has been witnessed worldwide

(Wada et al., 2010, 2012; Gleeson et al., 2012; Pokhrel et al., 2015), which has unsustainably impacted the long-term water supplies and impaired the health of many ecosystems (Wada et al., 2012; Gleeson et al., 2012).

Improving the representations of groundwater flow in Land Surface Models (LSMs), which serve as the land component of Earth System Models (ESMs), could help address important scientific questions (Clark et al., 2015). Groundwater movement in the critical zone, usually defined as the shallow groundwater (between 1-5 [m]), is important in modulating the terrestrial water-energy distribution and land-atmosphere interactions (Kollet and Maxwell, 2008; Condon and Maxwell, 2019; Fan, 2015). Long-term groundwater movement and storage variations are found highly influential in predicting the long-term water and energy partitions across different scales (Wang, 2012; Fang et al., 2016; Zhang et al., 2022). All these factors determine the important role that groundwater plays in regulating the eco-hydrological processes, especially in the groundwater-supplied ecosystems (Chui et al., 2011; Miguez-Macho and Fan, 2012; Subin et al., 2014; Vrettas and Fung, 2017; Fang et al., 2022). Moreover, the role of groundwater in regulating the water and energy balances at the land/atmosphere interface and how its feedback to climate change would affect ecosystem functioning at various spatial and temporal scales remain partly understood (Kløve et al., 2014; Clark et al., 2015). Lateral groundwater flow represents a critical process in representing groundwater dynamics. The magnitude of lateral groundwater flow is suggested to scale with grid resolution (Krakauer et al., 2014). With grid resolution less than 0.1° (∼10 [km]), lateral flow is comparable to the recharge rate and thus is non-negligible (Krakauer et al., 2014). Therefore, incorporating detailed representations of the lateral groundwater flow is important in LSMs regarding the surging interests in Hyper-resolution modeling at regional or global scales (Wood et al., 2011; Bierkens et al., 2015; Fan et al., 2019).

Despite the importance of groundwater systems in the terrestrial processes, the incorporation of lateral groundwater flow models in LSMs is just nascent. Nearly all LSMs that participated in the Climate Model Intercomparison Project version 6 (Eyring et al., 2016) ignored lateral groundwater flow. Most of these LSMs only simulated vertical soil water movement without lateral connections and parameterized the saturated groundwater dynamics with a lumped unconfined aquifer, e.g. CLM4.5 (Oleson et al., 2013), HiGW-MAT (Pokhrel et al., 2015). A few recent works have incorporated lateral groundwater flow within and across grids in LSMs. For example, Swenson et al. (2019) incorporated intra-grid saturated lateral groundwater flow into the Community Land Model (CLM) v5.0 at the hillslope scale. Recently, a number of inter-grid cell lateral groundwater flow models have been developed, which can be categorized into two major groups. The first group solves quasi-three dimensional (3D) groundwater flow that accounts for vertical soil water movement in the unsaturated zone and lateral groundwater flow in the saturated zone. For example, Zeng et al. (2018) coupled a lateral groundwater flow model in the saturated zone with CLMv4.5 (Oleson et al., 2013); Felfelani et al. (2021) extended the work of Swenson et al. (2019) to include inter-grid cell saturated lateral groundwater flow in CLM5.0 and applied the model at continental scale; Chaney et al. (2016, 2021) developed the HydroBlocks model by coupling the dynamic TOPMODEL, which used a kinematic wave approach and recently updated to Darcy flux, to represent the saturated lateral flow, with Noah-MP; H3D model (Troch et al., 2003; Paniconi et al., 2003; Hazenberg et al., 2015) couples a vertical 1-D soil column model with a pseudo-two dimension (2D) saturated groundwater model and was validated with a 3D Richards equation model (Richards, 1931); similarly, PAWS (Shen and Phanikumar, 2010; Shen et al., 2013) solves the saturation based ($\theta$-based) one dimensional (1D) Richards equation in the unsaturated zone

and 2D diffusive groundwater equation in the saturated zone, and was coupled with CLMv4.0 for solving land surface processes. The second group solves fully 3D groundwater flow in both the saturated and unsaturated zones. For example, ParFlow solves the variably saturated 3D Richards equation for both unsaturated and saturated groundwater and has a comprehensive representation of the surface and subsurface processes (Kollet and Maxwell, 2006, 2008; Maxwell, 2013). CLM-PFLOTRAN couples PFLOTRAN (Hammond and Lichtner, 2010) with CLM4.5 (Oleson et al., 2013), which simulates the 3D subsurface flow by solving the 3D variably saturated Richards equation and represents the land surface processes with CLM4.5 (Bisht et al., 2017). Similarly, Miura and Yoshimura (2020) developed the 3D Variably saturated groundwater model considering the storativity of groundwater, and validated the model for different idealized situations. However, the second group of models has not been applied at global scales but has only been applied to watershed-, regional-, and continental-scales studies due to high computational costs (Archfield et al., 2015).

The surging interest in applying hyper-resolution LSMs at continental or global scales motivated the development of more comprehensive and efficient representations of subsurface hydrology in LSMs (Archfield et al., 2015). Stemming from the Community Earth System Model (CESM) version 1_3_beta10 (Oleson et al., 2013), the Energy, Exascale, Earth System Model (E3SM) is a state-of-the-art ESM sponsored by the U.S. Department of Energy (Leung et al., 2020). The latest E3SM Land Model version 2.0 (ELMv2.0) solves the 1-D Richard equation in the unsaturated zone based on (Zeng and Decker, 2009) and parameterize the saturated groundwater process with a lumped unconfined aquifer. The goal of this study is to develop and validate a computationally efficient inter-grid cell lateral unsaturated and saturated groundwater flow model within ELMv2.0. Instead of solving the fully 3D Richards equation, the model solves a modified 1D $\theta$-based Richards equation including unsaturated and saturated zones, and considers the lateral groundwater flux as a source term. The developed model was first benchmarked against PFLOTRAN for three idealized hillslope planforms that included a convergent hillslope, divergent hillslope, and titled V-shape hillslope with variable saturated initial conditions. The model was next applied to the Little Washita Watershed (LWW) in the USA to assess its performance with field observations of soil moisture and soil temperature. The impacts of lateral flow on the surface energy fluxes were also evaluated.

## 2    Methods

### 2.1    Numerical formulation

#### 2.1.1    Lateral flow in unsaturated zone

The $\theta$-based Richards equation, which is often used to describe the water movement in the unsaturated zone, is given as

$$\frac{\partial \theta}{\partial t} = -\nabla \cdot \mathbf{q} - Q \tag{1}$$

where $\theta$ [mm$^3$ mm$^{-3}$] is the volumetric soil water content, $t$ [s] is time, $\mathbf{q}$ [mm s$^{-1}$] is the water flux, and $Q$ [mm$^3$ mm$^{-3}$ s$^{-1}$] is the sink of soil moisture. Use finite volume spatial discretization to rewrite the equation (1), and applying the finite

volume integral:

$$\frac{\partial}{\partial t} \int_{\Omega_n} \theta \, dV = -\int_{\Gamma_n} (\mathbf{q} \cdot \, dA) - \int_{\Omega_n} Q \, dV$$

$$\left( \frac{\partial \theta_n}{\partial t} \right) V_n = -\sum_{n'} (\mathbf{q}_{nn'} \cdot \mathbf{A}_{nn'}) - Q_n V_n \tag{2}$$

where $A_{nn'}$ $[m^2]$ is the common face area between the $n$-th and $n'$-th control volumes, $\Omega_n$ represents the $n$-th non-overlapping control volume with volume $V_n$, such that $\cup_i^n \Omega_i = \Omega$, $\Gamma_n$ represents the boundary surface of the $n$-th control volume. Applying semi-implicit time discretization and based on Taylor series expansion leads to:

$$\left( \frac{\Delta \theta_n^{t+1}}{\Delta t} \right) V_n = -\sum_{n'} (\mathbf{q}_{nn'}^{t+1} \cdot \mathbf{A}_{nn'}) - Q_n^{t+1} V_n$$

$$= -\sum_{n'} \left( \mathbf{q}_{nn'}^{t} + \frac{\partial \mathbf{q}_{nn'}^{t}}{\partial \theta_n} \Delta \theta_n^{t+1} + \frac{\partial \mathbf{q}_{nn'}^{t}}{\partial \theta_{n'}} \Delta \theta_{n'}^{t+1} \right) \cdot \mathbf{A}_{nn'} - Q_n^{t+1} V_n \tag{3}$$

where $\Delta \theta_n^{t+1}$ is the change in the volumetric liquid soil moisture over the time interval $\Delta t$.

In ELMv2.0, which is a 1D vertical model, a control volume at the $k$-th soil layer is only connected to soil layers above and below with no lateral connections to the $k$-th soil layer of the neighboring grid cell. The discretized equation (3) leads to a tridiagonal system of equations given as (Oleson et al., 2013):

$$a\Delta\theta_{k-1}^{t+1} + b\Delta\theta_k^{t+1} + c\Delta\theta_{k+1}^{t+1} = r \tag{4}$$

where:

$$a = -\left( \frac{\partial q_{k,k-1}^t}{\partial \theta_{k-1}^t} \right) \tag{5}$$

$$b = \left( \frac{\partial q_{k,k-1}^t}{\partial \theta_k^t} - \frac{\partial q_{k+1,k}^t}{\partial \theta_k^t} \right) - \frac{\Delta z_k}{\Delta t} \tag{6}$$

$$c = \left( \frac{\partial q_{k+1,k}^t}{\partial \theta_{k+1}^t} \right) \tag{7}$$

$$r = -\left( q_{k,k-1}^t - q_{k+1,k}^t + e_k \right) \tag{8}$$

where $q_{k,k-1}$ [mm s$^{-1}$] is the water flux between $(k-1)$-th and $k$-th soil layer, $\Delta z_k$ [mm] is the soil thickness of the $k$-th soil layer, and $e_k$ [mm s$^{-1}$] is the sink of water in the $k$-th soil layer.

The flux of water is given by Darcy's law as

$$q = -\kappa \frac{\partial(\psi + z)}{\partial z} \tag{9}$$

where $\kappa$ [mm s$^{-1}$] is the hydraulic conductivity and $\psi$ [mm] is the soil matric potential. The hydraulic conductivity and soil matric potential are modeled as the non-linear function of volumetric soil moisture (Clapp and Hornberger, 1978) as

$$\kappa = \Theta_{ice}\kappa_{sat}\left(\frac{\theta}{\theta_{sat}}\right)^{2B+3} \tag{10a}$$

$$\psi = \psi_{sat}\left(\frac{\theta}{\theta_{sat}}\right)^{-B} \tag{10b}$$

where $\kappa_{sat}$ [mm s$^{-1}$] is saturated hydraulic conductivity , $\psi_{sat}$ [mm] is saturated soil matric potential, $B$ [-] is a linear function of percentage clay and organic content (Oleson et al., 2013), and $\Theta_{ice}$ [-] is the ice impedance factor (Swenson et al., 2012). ELMv2.0 uses the modified form of Richards equation of Zeng and Decker (2009) that computes Darcy flux as

$$q = -\kappa\frac{\partial(\psi + z - C)}{\partial z} \tag{11}$$

where $C$ is a constant hydraulic potential above the water table, $z_\triangledown$, given as

$$C = \psi_E + z = \psi_{sat}\left[\frac{\theta_E(z)}{\theta_{sat}}\right]^{-B} + z \tag{12}$$

where $\psi_E$ [m] is the equilibrium soil matric potential and $\theta_E$ [mm$^3$ mm$^{-3}$] is the equilibrium volumetric soil water content. At the water table depth, $z = z_\triangledown$, the soil water content is $\theta_E(z_\triangledown) = \theta_{sat}$, thus $C = \psi_{sat} + z_\triangledown$. Substituting equation (12) in equation (11) leads to

$$q = -\kappa\frac{\partial(\psi - \psi_E)}{\partial z} \tag{13}$$

In this work, we modified the ELMv2.0 tridiagonal system given by equation (4) to include unsaturated lateral flux between $g$-th and $g'$-th grid cell for the $k$-th soil layer (Figure 1). The ELM with the newly developed lateral flow model is hereafter abbreviated as ELM$_{lat}$. Specifically, the equation (8) is modified to account for unsaturated lateral flux, which uses an explicit time integration scheme, and yields

$$r = -\left(q^t_{k,k-1} - q^t_{k+1,k} + e_k\right) + \sum_{g'}\left(q^{ulat,}_{g_k,g'_k}\right)^t\left(\frac{A^{gg'}_k}{V^g_k}\right) \tag{14}$$

where $q^{ulat}_{g_k,g'_k}$ is lateral flux between grid cell $g$ and its neighbor cells $g'$ for the $k$-th soil layer.

Following Childs (1971), Henderson and Wooding (1964) and Maxwell (2013), we adapt the grid alignment in ELMv2.0 with parallelogram grid cells to better represent the real-world terrain. The adaption is illustrated in Figure 1 for a x-z transect with a uniform $\Delta z$. In this setup, the lateral Darcy flux in unsaturated zone is modified to follow the grid alignment in Figure 1 (a) (Childs, 1971; Celia et al., 1990; Maxwell, 2013):

$$q^{ulat}_{g_k,g'_k} = -\kappa_x\left(\frac{\partial(\psi + z)}{\partial x}cos\theta_x + sin\theta_x\right) \tag{15}$$

where $\theta_x$ is the angle of slope at the horizontal (x) direction between two neighbouring cells. It should be noted that the vertical flux may not be perpendicular to the land surface given the parallelogram grid alignment. By performing the dot

product between flux and area for the vertical direction in Equation (3), the grid horizontal surface area is multiplied by the cosine of $\theta_z$, which can be expressed as:

$$A'_s = A_s cos\theta_z \tag{16}$$

where $A_s$ is the grid horizontal surface area, $\theta_z$ is the angle between the normal vector of the cell surface at cell center and the vertical direction (z). The lateral Darcy flux at y-direction adopts the same computation method as equation (15) .

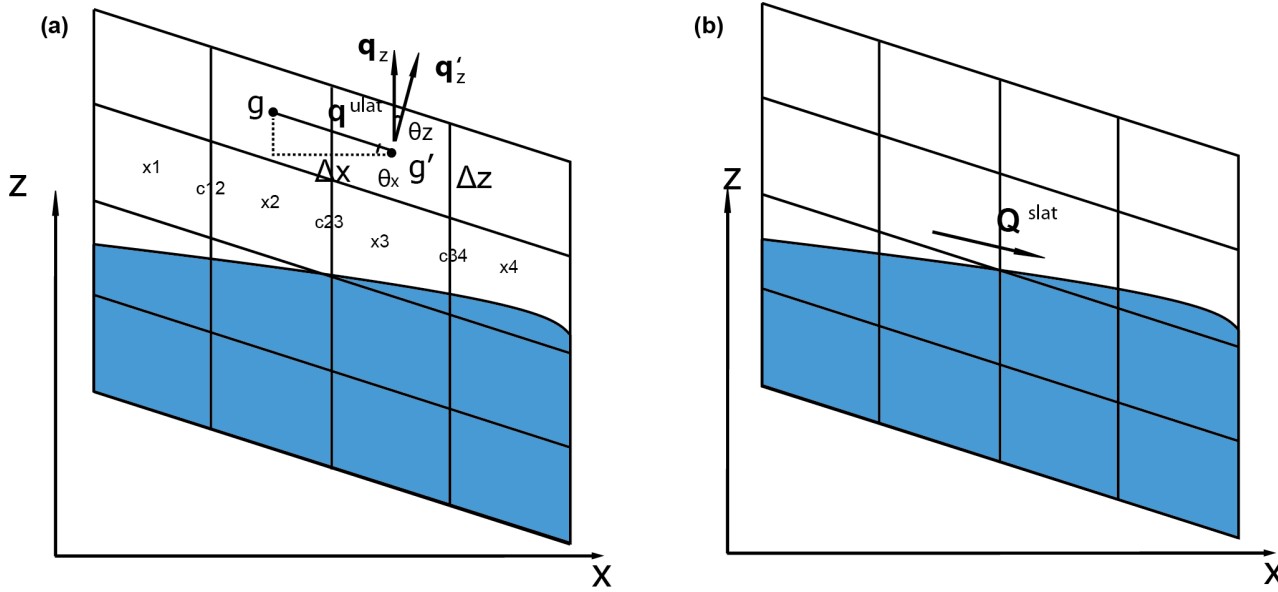

**Figure 1.** Sketches of the terrain following grid formulations, with illustrations of (a) unsaturated and (b) saturated subsurface lateral flow calculation. $\theta x$ is the local angel of slope at the horizontal (x) direction, and $\theta z$ is the angle between the normal vector of the cell surface and the vertical direction (z)

## 2.2   Lateral flow in saturated zone

The lateral flow in the saturated zone, $Q^{slat}$ $[m^3/s]$, is also modeled using Darcy's law while taking the soil matric potential gradient as the groundwater hydraulic head gradient. In such a manner, the volumetric lateral groundwater flow can be given as:

$$Q^{slat} = -wT \left( \frac{\partial z_{\nabla}}{\partial x} cos\theta_x + sin\theta_x \right) \tag{17}$$

where $w$ [m] is the width of the grid cell, and $T$ [m²/s] is the transmissivity. The transmissivity is calculated following the method used in Fan et al. (2007), see Appendix A. The $Q^{slat}$ is computed across inter-grid cell using the same grid cell

connections as used for the $Q^{slat}$. The water table depth for each soil column within a grid cell is thereafter adjusted based on $Q^{slat}$. Specifically, the $Q^{slat}$ increases/decreases the soil water content of the soil layer where the groundwater table is currently located. The soil moisture is updated while ensuring the soil moisture remains below/above the maximum/minimum allowable value. If the change in the soil moisture is less than the $Q^{slat}$, the soil moisture of layers above/below is recursively increased/decreased till the total change in soil moisture in all updated soil layers matches $Q^{slat}$. In ELMv2.0, the groundwater table is adjusted by a conceptual recharge flux which depends on whether the water table is within or below the soil column. The approach by explicitly subtracting the hydrostatic equilibrium soil moisture distribution from the Richards equation resolved the numerical deficiencies when the water table is within the soil columns while using $\theta$-based Richards equation (Zeng and Decker, 2009; Yu et al., 2014). In $\text{ELM}_{lat}$, the water table is assumed within the soil columns, such that the conceptual recharge flux is no longer employed. The groundwater table depth is computed based on the soil water content following the $\theta$-based water table method in CLM5.0 (Figure 2). A no flux boundary condition is applied to the last hydrologically active soil layer. Following ELMv2.0, the saturated zone is depleted by a drainage flux, i.e. the subsurface runoff, that is computed based on the SIMTOP scheme of Niu et al. (2007), with modifications to account for frozen soils (Oleson et al., 2013).

## 2.3 Model benchmarking against PFLOTRAN for idealized hillslopes

Three idealized hillslopes that included a convergent hillslope (CH), divergent hillslope (DH), and titled V-shape hillslope (VH) with variable saturated initial conditions are used to validate $\text{ELM}_{lat}$ by benchmarking against a 3D subsurface flow and transport model, PFLOTRAN.

### 2.3.1 Idealized hillslope geometries

The CH/DH problems have been widely used to test the newly developed lateral groundwater flow model, e.g. (Troch et al., 2003; An et al., 2010; Hazenberg et al., 2015). Different from ELMv2.0 which uses uniform grid cells, the surface area of CH/DH grid cells change along the slope. The surfaces of CH/DH are curved by the sinusoidal function (Figure 3):

$$z(x,y) = 10\sin\left(\frac{x\pi}{100} - \frac{\pi}{2}\right) + \frac{y}{50} \tag{18}$$

where x, y, and z are the coordinates of the cell vertices. For the CH, the total width along the y-direction linearly shrinks from 100 m at hill top to 80 m at hill bottom; whereas for the DH, it linearly expands from 80 m at hill top to 100 m at hill bottom (Figure 3). The total width along the x-direction is 100 m for both CH and DH.

The tilted V-catchment hillslope is another popular benchmark problem for validating groundwater flow in land surface models (Park et al., 2009; Sulis et al., 2011; Maxwell et al., 2014). As shown in Figure 3 (c), the V-shape catchment is formed by the union of two symmetrically inclined planar surfaces (80 m $\times$ 100 m) on the sides connected by a channel in the middle (20 m $\times$ 100 m). The two planar surfaces are inclined with slopes of $\pm$ 0.05 and 0.02 at x and y directions, respectively. The channel is inclined following the slope of 0.02 at the y direction.

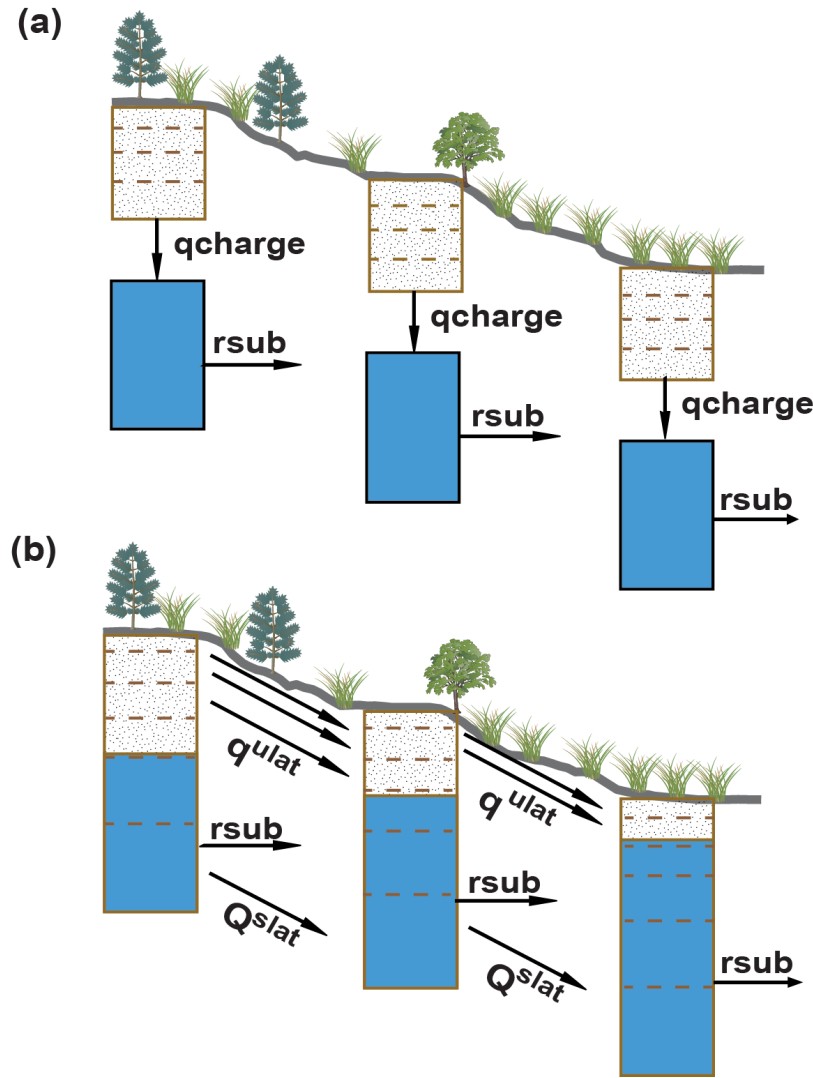

**Figure 2.** Schematic illustration of the subsurface flow implementation in unsaturated and saturated zone from (a) ELMv2.0 to (b) ELM$_{lat}$; qcharge is the conceptual recharge flux used to update the groundwater table, $q^{ulat}$ is the lateral flux in unsaturated zone, $q^{slat}$ is the lateral flux in saturated zone, rsub is the subsurface runoff

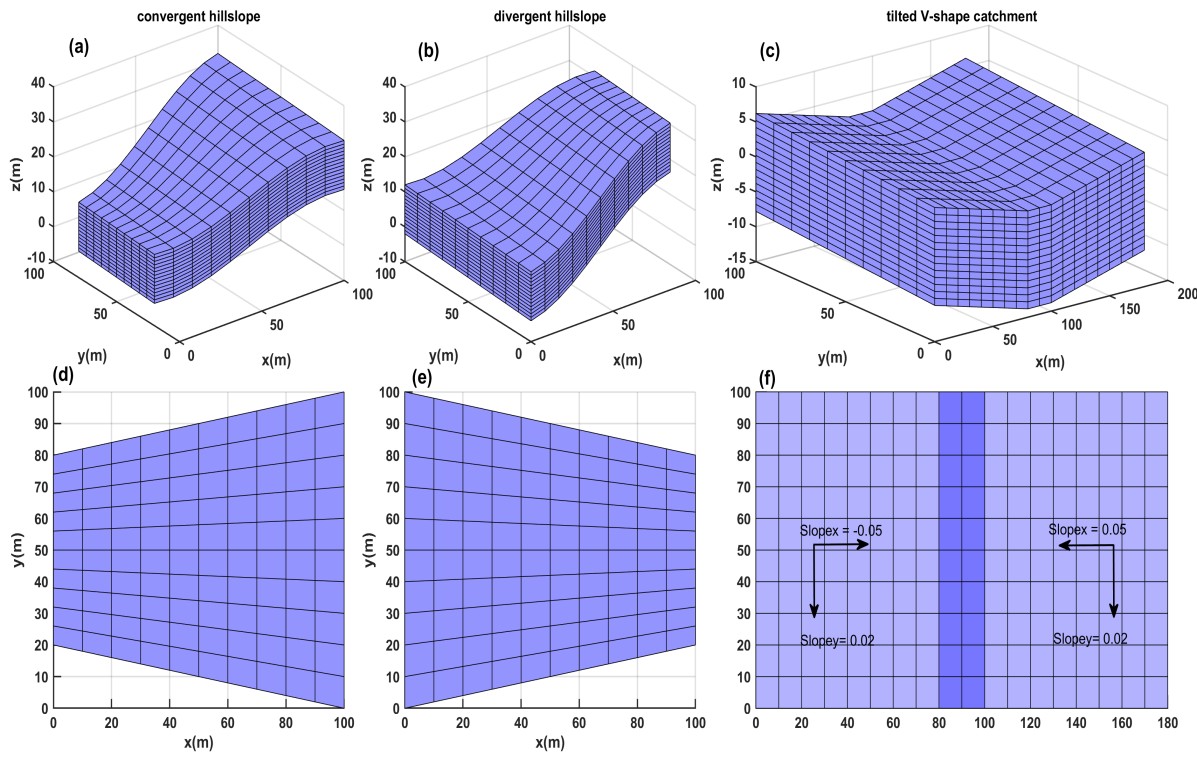

**Figure 3.** Geometries of the three benchmark test problems. (a), (b), (c) are 3D views, and (d), (e), (f) are x-y views of the three benchmark test problems, respectively. $S_x$ and $S_y$ are slopes along the x and y directions, respectively.

### 2.3.2 Model setup

By default, the soil columns of ELMv2.0 are vertically discretized into 15 soil layers of exponentially varying soil thicknesses reaching a depth of 42.1 [m]. Only the first 10 soil layers are hydrologically active and occupy the top 3.8 [m] of a soil column. For the model benchmarking, we modified the default model setup by allowing all 15 soil layers to be hydrological active and have a uniform soil thickness of 1 [m] to be consistent with the setup for PFLOTRAN simulations. The soil columns were horizontally discretized to a $10 \times 10$ mesh with a grid resolution of 10 [m] in the x direction and spatially changing resolution in the y direction for CH and DH. For VH, the soil columns were discretized horizontally with a uniform grid resolution of 10 [m] in both x and y directions, which produced a mesh of $18 \times 10$. A homogeneous soil texture was used with a porosity of 0.467. The soil water retention properties in ELMv2.0 were described using the Clapp-Hornberger formula (Clapp and Hornberger, 1978) while in PFLOTRAN they were described using the Brooks Corey's formula (Brooks, 1965). We matched the soil water retention property parameters in PFLOTRAN with ELM. The translation of soil parameters between the two models is described in Appendix B. The anisotropic ratio for the horizontal to vertical hydraulic conductivity was set as 1.0 for the CH

and DH cases ($K_x = K_y = K_z$) and 10.0 for the VH case ($K_x = K_y = 10.0K_z$). We used higher anisotropic ratio for the VH case to accelerate the lateral water movement for visualizing more evident soil moisture and groundwater dynamics. In order to evaluate the model sensitivity in response to the anisotropic ratio, we performed additional simulations using the anisotropic ratio of 10.0 for the CH and DH cases and 1.0 for the VH case. The same boundary and initial conditions were applied for the three benchmark problems. A no-flow boundary condition was applied on all sides of the domain. Hydrostatic pressure was used as the initial condition with the groundwater table depth (WTD) set at 7 [m] deep from the top surface. ELM$_{lat}$ and PFLOTRAN simulations were performed for 80 days for all three benchmark problems and results were compared at the end of the simulation. The performance of soil moisture dynamics was evaluated using the mean absolute error (MAE) and root mean square error (RMSE) between ELM$_{lat}$ and PFLOTRAN, while the performance of groundwater was directly evaluated by the difference between ELM$_{lat}$ and PFLOTRAN simulations.

## 2.4 Model application

### 2.4.1 Study region

ELM$_{lat}$ is applied to study the role of lateral subsurface flow on terrestrial processes in the Little Washita Watershed (LWW), which is located in southwestern Oklahoma, USA (Figure 4). The LWW has a drainage area of $\approx$611 km$^2$ and is one of the seven selected experimental watersheds jointly administrated by the U.S. Department of Agriculture (USDA) and U.S. Environmental Protection Agency for a variety of hydrologic research projects. The LWW has a subhumid climate with annual precipitation of approximately 760 [mm] and annual temperature of approximately 16 [°C]. The elevation of LWW ranges from approximately 320 [m] to 460 [m], and the slope ranges from 0° to 3° (Figure 4). Grassland is the dominant land cover of LWW, with small portions of land occupied with crops, shrubs, and deciduous trees. The mean annual Leaf Aare Index (LAI) is within the range of 0.5 - 3.0 [m$^2$/m$^2$]. Soil textures of LWW are composed of sand, loam, and silty loam (Allen and Naney, 1991). LWW has a relatively higher soil content of sand and organic matter in the northwestern part and southeastern part; several spots in the southern region have higher clay content (Figure 4a-c).

### 2.4.2 Model configuration and data

The model domain covered a 35 [km] × 50 [km] area encompassing the LWW and was laterally discretized at 1 [km] × 1 [km]. All 15 soil layers were set as hydrologically active. Results from Fan et al. (2013) show the maximum WTD could reach up to 60 [m] deep in this region, which is deeper than the depth of ELM's 15-layer soil column. Therefore, we modified the soil thickness of the last two soil layers by adding 20 [m] each to their default values in ELMv2.0 while keeping the discretization of the other 13 soil layers unchanged. The total soil thickness could reach $\approx$ 82 [m]. The initial WTD was spatially uniformly set at 8 [m] deep below the top surface of the domain. The hourly NLDAS data (https://ldas.gsfc.nasa.gov/nldas/v2/forcing) was used as the climate forcing which has a spatial resolution of 1/8th-degree grid. We performed 100-year simulations with ELM$_{lat}$ and ELMv2.0 while cycling the atmospheric forcing data of 2008 to allow sufficient time for the model spin-up. The model results from the last year of the spin-up were used for analysis.

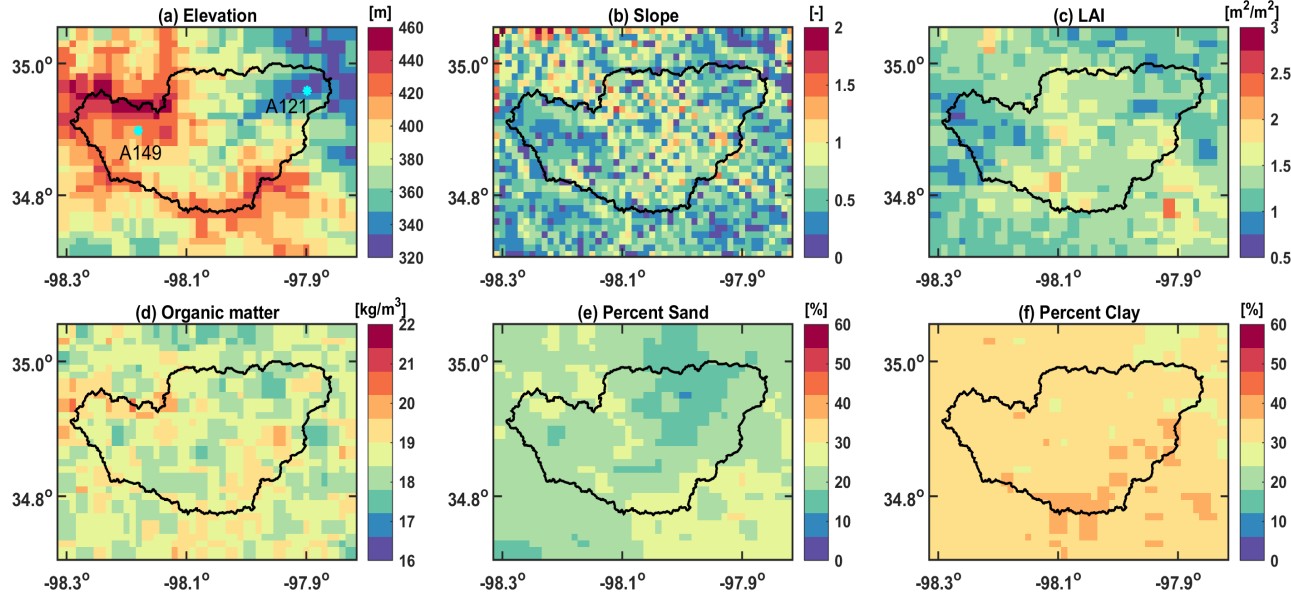

**Figure 4.** Spatial distributions of (a) elevation, (b) slope, (c) mean monthly leaf area index, (d) organic matter density, and (e–f) percentages of sand and clay, respectively, over the study area at a resolution of 1 km.

The recently developed 1 [km] surface data of (Li et al., 2023) was used in this study, which was generated using multiple high-resolution, sub-kilometer data sets including vegetation, soil, and topography-related variables (Table 1). The soil, elevation and LAI datasets were first aggregated to 1 [km] using the area-weighted average method, and the land cover data was aggregated to 1 km using the majority resampling method. Specifically, MODIS 500 [m] land cover was at year 2005 obtained
from the Google Earth Engine (Gorelick et al., 2017, GEE;). Following Ke et al. (2012), the LC_Type 5 class of MCD12Q1 v6 land cover product (Friedl and Sulla-Menashe, 2019) was used to determine the lake, urban, glacier, and vegetation plant functional types (PFTs). These PFTs were further classified into tropical, temperate, and boreal sub-types based on the rules presented in Bonan et al. (2002) and using meteorological data from WorldClim V1 (Hijmans et al., 2005). The monthly climatology LAI was derived from the 4-day MCD15A3H V6.1 (Myneni et al., 2021) over 2003–2020 using GEE. Then based
on the method in Zeng et al. (2002), we calculated the monthly climatology stem area index (SAI) from the LAI data. For the soil data, we used the Soilgrid v2 data with an original resolution of 250 m (Poggio et al., 2021). Soilgrid is generated based on machine learning using multiple data sources of soil profiles and remote sensing data (Hengl et al., 2017). The percent clay, percent sand, and soil organic matter at multiple depths were processed for ELM. The 90 m digital elevation from the Shuttle Radar Topography Mission (Jarvis et al., 2008) was used to derive topography-related parameters in ELM, including
the standard deviation of elevation and slope. These 0.01 deg datasets were processed using GEE based on the original 90 m elevation.

**Table 1.** Specifications of high-resolution data sets used in this study.

| Type | Parameter | Spatial resolution | Temporal resolution | Data source | Reference |
|---|---|---|---|---|---|
| Vegetation | Vegetation type | 500 m | yearly | MODIS MCD 12Q1 V6 | Friedl and Sulla-Menashe (2019) |
| | leaf area index | 500 m | 2003–2020, 4-day | MODIS MCD15A3H V6.1 | Myneni et al. (2021) |
| | Stem area index | 500 m | 2003–2020, 4-day | Calculated | |
| Soil | Percent sand | 250 m | Static | Soilgrid v2 | Poggio et al. (2021) |
| | Percent clay | 250 m | Static | Soilgrid v2 | Poggio et al. (2021) |
| | Organic matter | 250 m | Static | Soilgrid v2 | Poggio et al. (2021) |
| Topography | Elevation | 90 m | Static | SRTM v4 | (Jarvis et al., 2008) |

The anisotropic ratio of the hydraulic conductivity has a close relationship with soil property, because the platy mineral form and the low permeability of the clay content has a strong effect on the anistropy (Fan et al., 2007). The anisotropic ratio is set as 10 ($K_x = K_y = 10K_z$) referred from table 2 of Fan et al. (2007) based on the primary soil property of LWW. We used the WTD map (Fan et al., 2013), which is hereafter referred to as Fan2013, with a resolution of 30 arc-second ($\approx$1 km) to validate the simulated WTD.

In ELMv2.0, the subsurface drainage flux ($q_d$) which is calculated dependent on the water table depth, Niu et al. (2005), plays a role in regulating the long term groundwater table.

$$q_d = q_{d,max} \exp\left(-f_d z_\nabla\right) \tag{19}$$

where $q_{d,max}$ [kg m$^{-2}$ s$^{-1}$] is the maximum drainage flux that depends on the local slope of a grid cell, $f_d$ [m$^{-1}$] is the subsurface drainage parameter. To improve the WTD simulations, the $f_d$ values were calibrated by performing an ensemble of simulations with $f_d$ values of 0.1, 0.5, 1.0, 2.5, 5.0, 10.0 [m$^{-1}$], respectively. A 100 year simulation was performed for each $f_d$ value, and the results from the last year of the simulation were employed for analysis. Following the approach of Bisht et al. (2018), a nonlinear functional relationship was established between the simulated WTD and $f_d$ values at each grid cell. The Fan2013 dataset was next employed to estimate an optimal $f_d$ based on the nonlinear WTD-$f_d$ relationship. The

impacts of lateral flow on the model performance were assessed by comparing the soil moisture and energy fluxes including soil temperature, latent and sensible heat flux, with ELMv2.0 simulation. Site-level observations of soil moisture and soil temperature were collected from the ARS Micronetwork (https://ars.mesonet.org/) which is operated and maintained by the USDA. The magnitude of unsaturated lateral flux against lateral flux was evaluated for both the idealized problems and the LWW. An experiment simulation on LWW was performed to evaluate the effects of unsaturated lateral flux on the energy fluxes by closing the unsaturated lateral flux while keeping the saturated lateral flux.

## 3 Results and discussion

### 3.1 Evaluation of simulations for the benchmark problems

$ELM_{lat}$ can accurately reproduce the evolution of vertical soil moisture profile simulated by PFLOTRAN for all three benchmark problems (Figure 5). The model correctly simulates the drying out of uphill soil columns (Figure 5a-c) and the wetting up of downhill soil columns (Figure 5d-f) for all the three hillslopes. In the VH case, the soil moisture shifted quickly during $20 - 50$ [day] but slowed down during $50 - 80$ [day], whereas the rate of soil moisture change is nearly steady for the CH and DH cases during the two periods. During the period of 50 - 80 [day], the VH case is much closer to a hydrostatic condition than the other two cases. The largest discrepancies in the simulated soil moisture are in soil layers that are closer to the water table, and these discrepancies could be explained by the differences in the model structure (Figure A1 and Figure A2). PFLOTRAN solves the variably saturated subsurface flow equation, which is applicable for both the unsaturated and saturated zones, while $ELM_{lat}$ has two separate flow models for the unsaturated and saturated zones. Thus, the difference between the two models is expected to be the largest near the unsaturated-saturated transition zone.

At the end of the simulation, the soil moisture was redistributed and transported following the surface topography via the lateral flow (Figure 6a-c). The most downhill columns of CH and DH are approaching saturated, as shown in Figure 6d-f. The difference between the two models was smaller than $\pm 2\%$ for the top ten layers on the 80-th simulation day, which provides confidence in the ability of $ELM_{lat}$ to simulate lateral unsaturated soil moisture dynamics. During the simulation period, the MAEs in $ELM_{lat}$ simulated soil moisture for all cells in the top 10 soil layers remain within $1\% \pm 3\%$, Figure A3(a,c,e), and the RMSE for all three test problems remain within 0.04, Figure A3(b,d,f).

At the end of the simulation, the simulated groundwater table evolves from the spatially uniform initial depth of 7 [m] below the top surface to a spatially varying depth that is inversely related to the surface elevation (Figure 7a-c). . In $ELM_{lat}$, the lateral and vertical flow in unsaturated zone is carried out simultaneously in the same tridiagonal equation, while the saturated/unsaturated lateral flow is calculated sequentially. In PFLOTRAN, the saturated and unsaturated flow is simulated with a single variably saturated flow equation. Additionally, it should be noted that the prognostic variable in PFLOTRAN is the soil water pressure and the simulated WTD is diagnosed by linearly interpolating the vertical pressure profile for each soil column. Despite those differences, $ELM_{lat}$ simulated WTD is comparable to that of PFLOTRAN with differences within $\pm 0.2$ m throughout the simulation period (Figure A4).

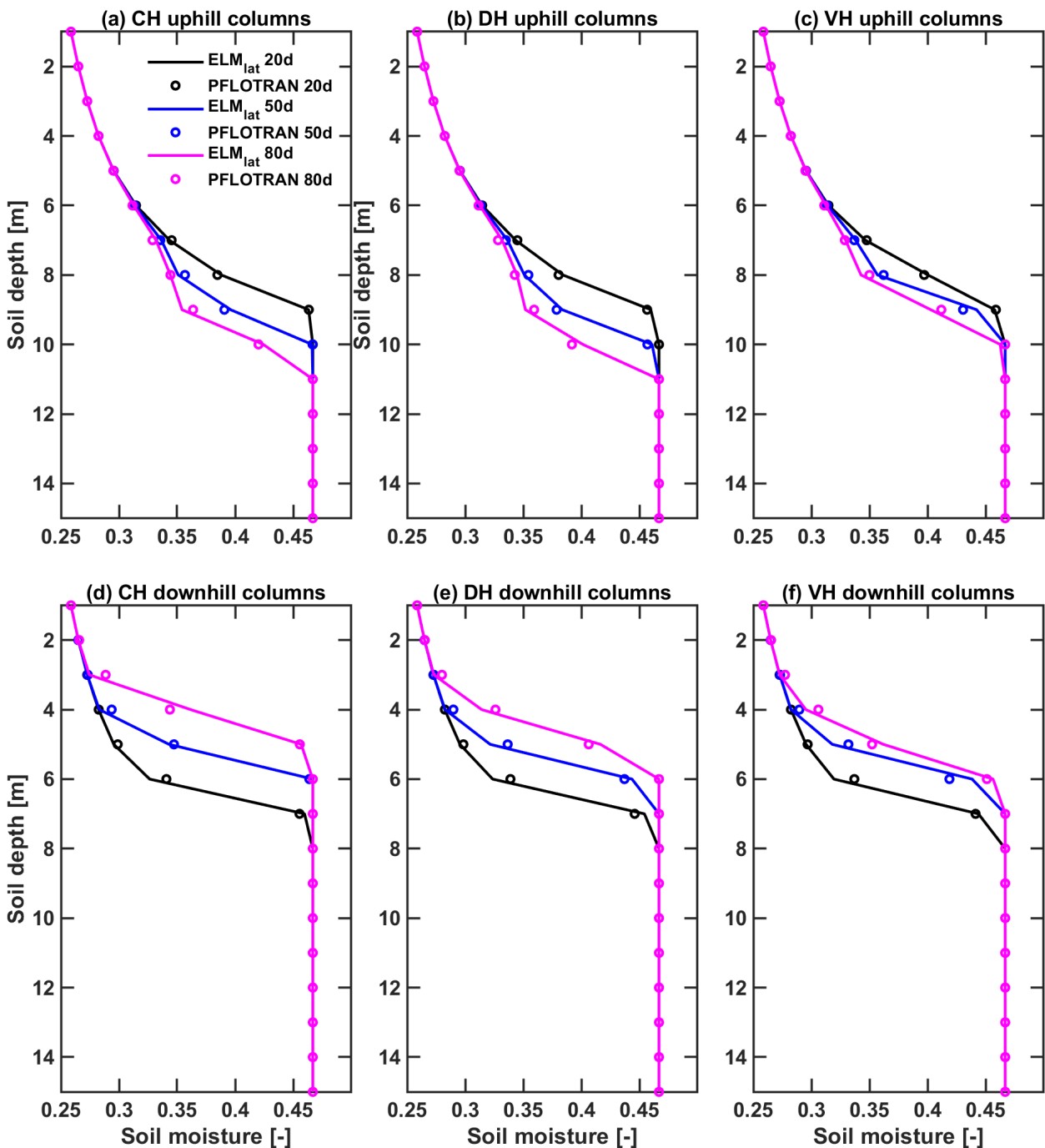

**Figure 5.** The comparison of average soil moisture profile for the (a-c) uphill and (d-f) downhill columns by $ELM_{lat}$ (circles) and PFLO-TRAN (lines) for the (a,d) convergent hillslope, (b,e) divergent hillslope, and (c,f) V-shaped hillslope. The results of the models are presented at the end of 20th, 50th, and 80th simulation day, respectively.

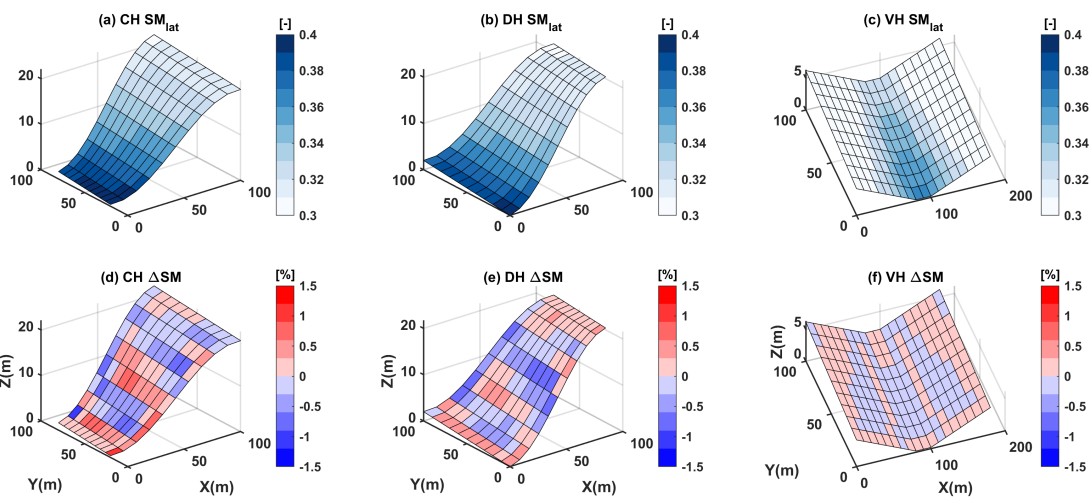

**Figure 6.** ELM$_{lat}$ simulated average soil moisture for the top 10 layers at $80th$ day for (a) convergent hillslope, (b) divergent hillslope, (c) V-shape hillslope, and the differences with PFLOTRAN for (d) convergent hillslope, (e) divergent hillslope, (f) V-shape hillslope, respectively.

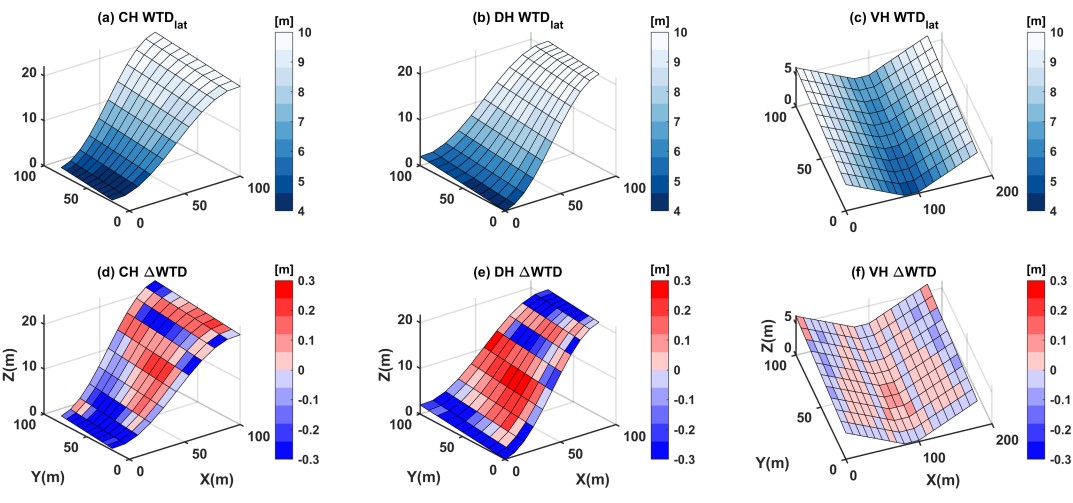

**Figure 7.** ELM$_{lat}$ simulated water table depth at 80th day for (a) convergent hillslope, (b) divergent hillslope, (c) V-shape hillslope, and the differences with PFLOTRAN for (d) convergent hillslope, (e) divergent hillslope, (f) V-shape hillslope, respectively.

For the CH and DH cases, with higher anisotropic ratio ($K_x = K_y = 10K_z$) and therefore larger lateral hydraulic conductivity and flow rate, the drying-down on the uphills and wetting-up on the down hills went much faster (Figure A5 a,b,d,e). The RMSE values of simulated soil moisture between $ELM_{lat}$ and PFLOTRAN for the top ten soil layers for CH and DH are 0.016 0.017, respectively. These RMSE values are larger than the case with anisotropic ratio of 1.0 that were 0.011 and 0.0125 for CH and DH, respectively. For the VH cases with anisotropic ratio of 1.0, water moves slowly since the average slope is low. The simulated results at the top ten layers for the VH case are very close between the two models with RMSE value of 0.003, while the RMSE value for the higher anisotropic ratio is 0.010. Overall, the above evaluation results demonstrates $ELM_{lat}$ can accurately represent vertical and lateral transport of soil moisture and groundwater.

Relative to lateral unsaturated groundwater flow, the lateral saturated groundwater flow is significantly larger. For all the three benchmark cases, the magnitude of the unsaturated flow was $\sim 10^{-6}$ [mm/s], while the magnitude of saturated flow is at $\sim 10^{-3}$ [mm/s], as shown in Figure 8. Hydraulic conductivity is nonlinearly dependent on soil saturation conditions and varies significantly with soil properties (Anderson et al., 2015). The scale difference of the hydraulic conductivity between unsaturated flow ($\sim 10^{-10}$ [m/s]) and saturated flow ($\sim 10^{-7}$ [m/s]) is the primary reason for the magnitude difference between unsaturated and saturated lateral flux.

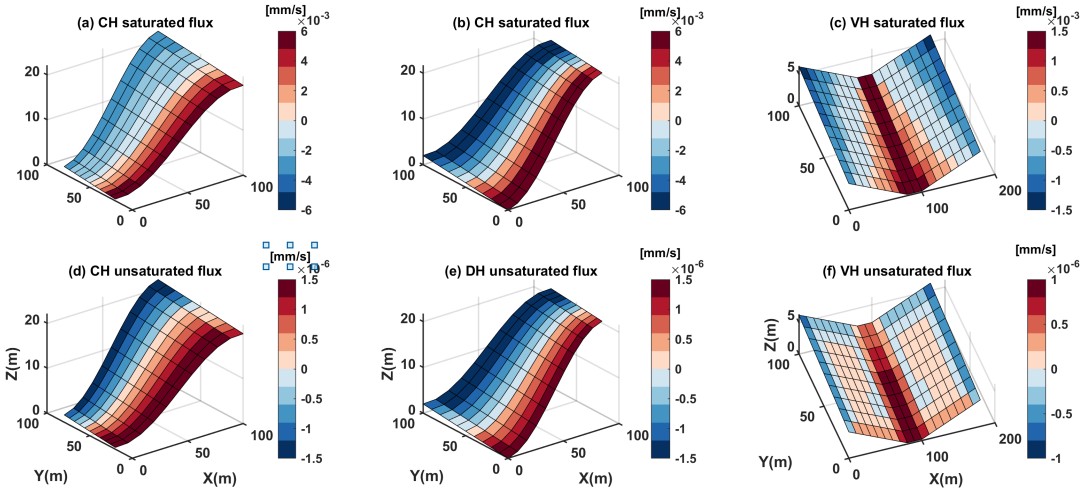

**Figure 8.** The magnitude of saturated lateral flux vs. unsaturated lateral flux for the three idealized problems: (a),(b),(c) are the magnitude of saturated lateral fluxes for CH,DH and VH problems and (d),(e),(f) are the magnitude of unsaturated lateral flux for CH,DH and VH problems.

### 3.2 Evaluation of simulations over LWW

Using the nonlinear relationship of WTD-$f_d$, an optimized $f_d$ was obtained for 99% of grid cells. Figure A6 (a) shows an example of the WTD-$f_d$ relationship and the spatial distribution of the calibrated $f_d$ value is shown in Figure A6 (b). With

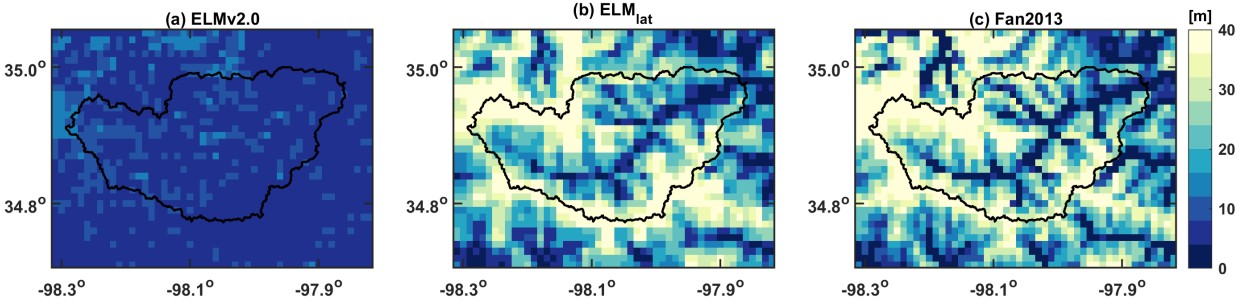

**Figure 9.** Annual average groundwater table depth (WTD) simulated by (a) ELMv2.0 and (b) ELM$_{lat}$, along with (c) the Fan2013 dataset

optimized $f_d$ values, the simulated WTD dynamics in LWW simulated by $ELM_{lat}$ is significantly improved as compared to the ELMv2.0 results (Figure 9). Simulated WTD by ELM$_{lat}$ showed a strong correlation ($r^2$ = 0.85) with surface topography, which is consistent with the Fan2013 results ($r^2$ = 0.87), while ELMv2.0-simulated WTD has no obvious spatial variations (Figure 9). Relative to ELMv2.0, including the lateral groundwater flow increased the baseflow of the runoff, as shown in Figure A7. Redistributed WTD increased the subsurface runoff while the timing of the peak runoff was not changed.

ELM$_{lat}$ simulated generally lower soil temperature (ST) and sensible heat flux (SH), but higher latent heat flux (LH) than ELMv2.0 for most of the grid cells (Figure 10). The spatial annual ST showed a gradually increasing trend from north to south. The LH and SH showed the opposite spatial pattern, where LH is higher the SH is lower. The effects of WTD changes on the energy fluxes were more pronounced at low elevation cells, especially at the stream and its surrounding cells. The delivery of the groundwater through the lateral flow to the valleys supported higher LH while reducing the SH compared with ELMv2.0

which has little spatial WTD variations.

   Both ELM$_{lat}$ and ELMv2.0 were able to capture the major fluctuations and wetting/drying cycles of soil moisture (SM) comparing with observations at the two stations (Figure 11). However, the dry-down rates of soil moisture results are not perfectly captured by both ELM$_{lat}$ and ELMv2.0. It should be noted that the simulated results represent the model behavior of the whole 1 [km] grid, while the measurements are taken at a single point. It is probable that the soil parameters for the 1

[km] grid could not accurately represent the soil property of the single point. Station A121 is located at the lowland area of this catchment with an elevation of 343 [m] while station A149 is located in the high land area with an elevation of 420 m (Figure 4(a)). At station A121, ELM$_{lat}$ simulated higher soil moisture, with the annual mean 0.013 higher at 5cm and 0.021 higher at 25 [cm] than ELMv2.0 results. At station A149, ELM$_{lat}$ simulated slightly lower soil moisture, with annual mean 0.007 lower at 5cm and 0.009 lower than ELMv2.0 results.

For the soil temperature simulations, summer months show a larger change due to lateral subsurface flow, therefore we focused our analysis on the summer temperature (Figure 12). Relative to ELMv2.0, ELM$_{lat}$ simulated cooler summer temperature at both 5 [cm] (mean $\Delta$ ST = -0.065 $^\circ C$) and 30 [cm] depths (mean $\Delta$ ST = -0.062 $^\circ C$) at station A121. In contrast, at station A149, ELM$_{lat}$ simulated slightly higher summer temperatures at soil depths of 5 cm (mean $\Delta$ ST = 0.017 $^\circ C$) and 30 [cm] (mean $\Delta$ ST = 0.019 $^\circ C$). The differences in summer temperature between ELM$_{lat}$ and ELMv2.0 were consistent with

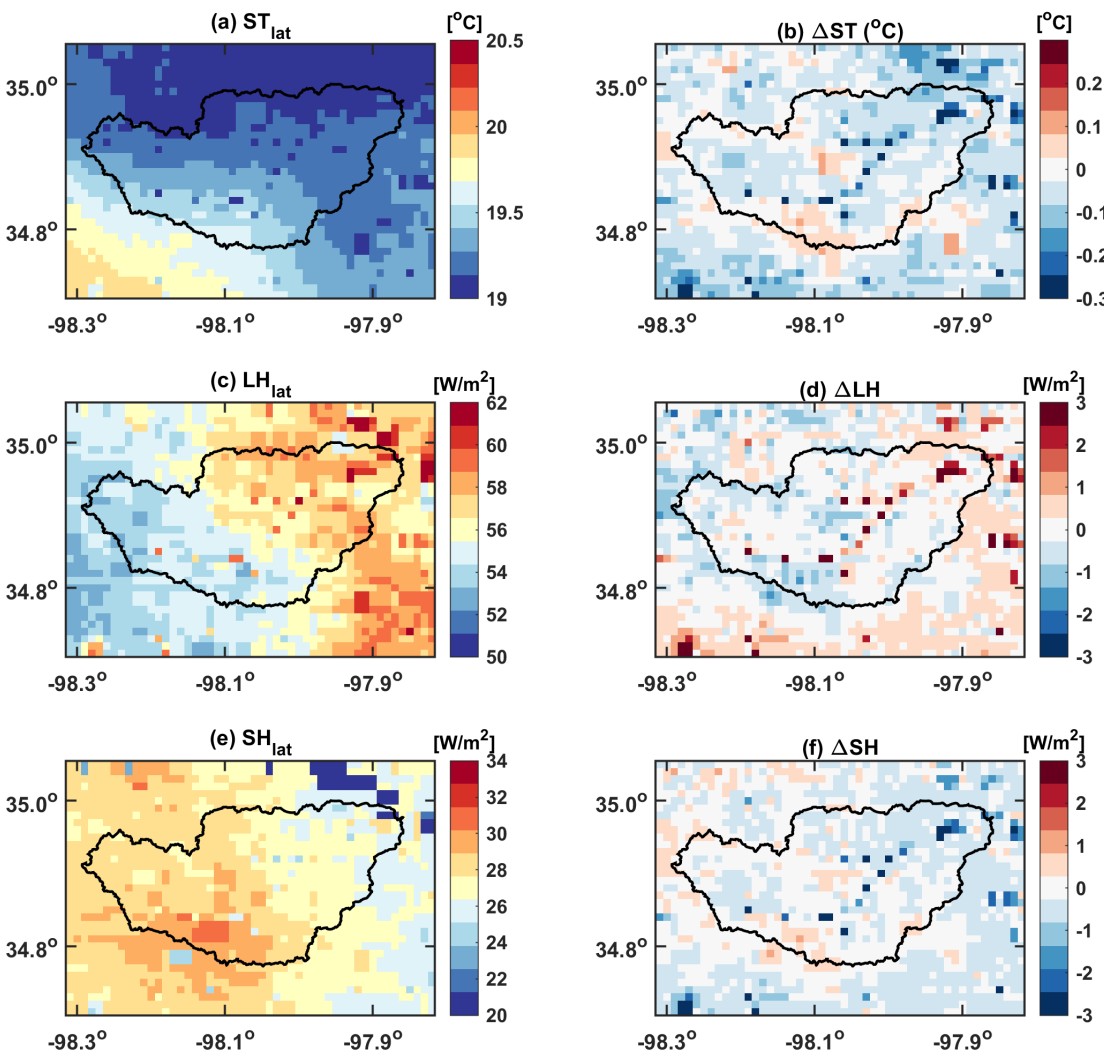

**Figure 10.** ELM$_{lat}$ simulated annual (a) top layer soil temperature [$^oC$], (c) latent heat flux [W/m$^2$] and (e)sensible heat flux [W/m$^2$]; (b), (d), (f) are the differences between ELM$_{lat}$ and ELMv2.0 simulations for the three energy items, respectively

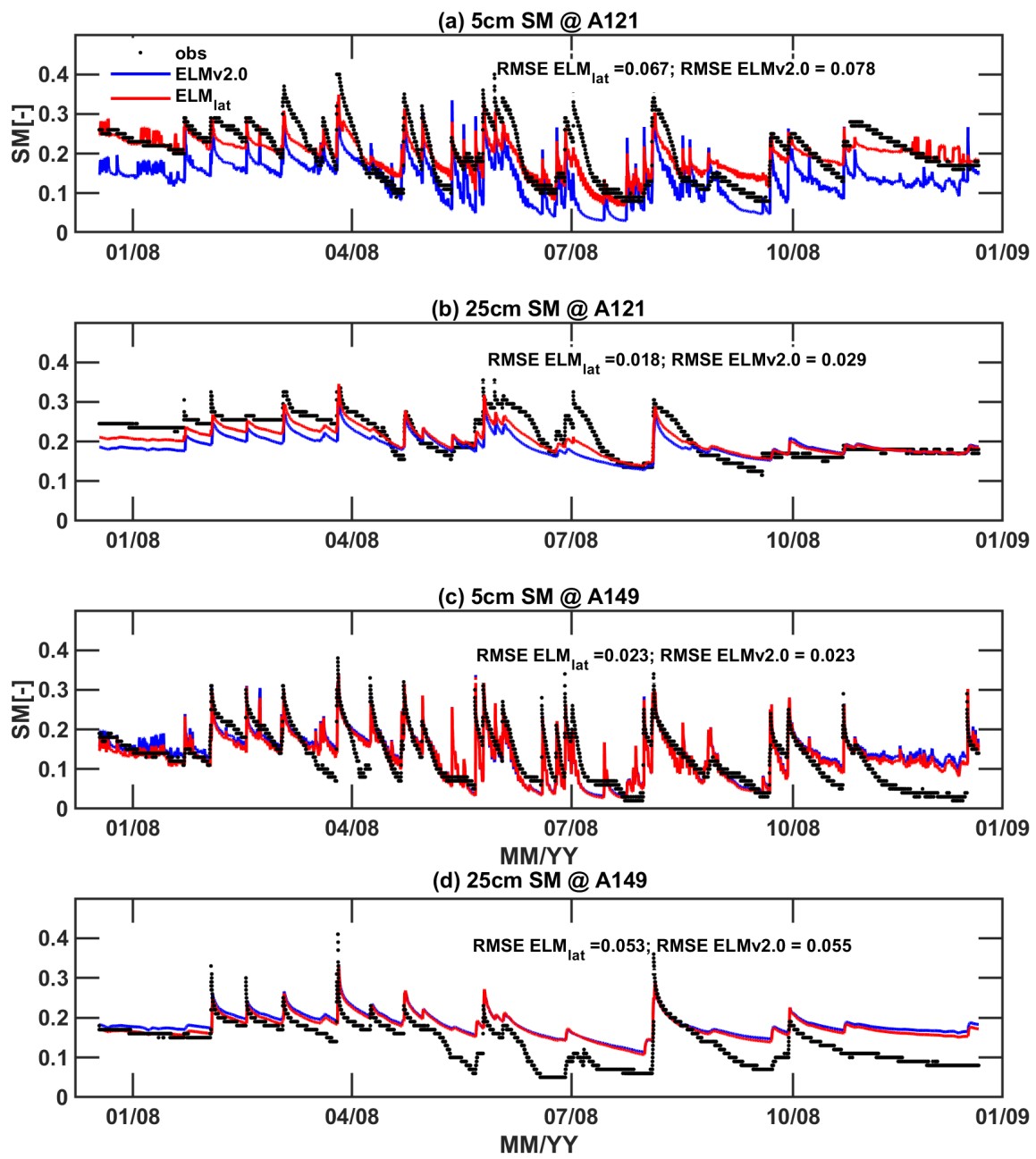

**Figure 11.** comparisons of simulated hourly soil moisture at depths of (a) 5 cm and (b) 25 cm of a low land station (A121) and at depths of (c) 5 cm and (d) 25 cm of a high land station (A149) between $ELM_{lat}$ and ELMv2.0

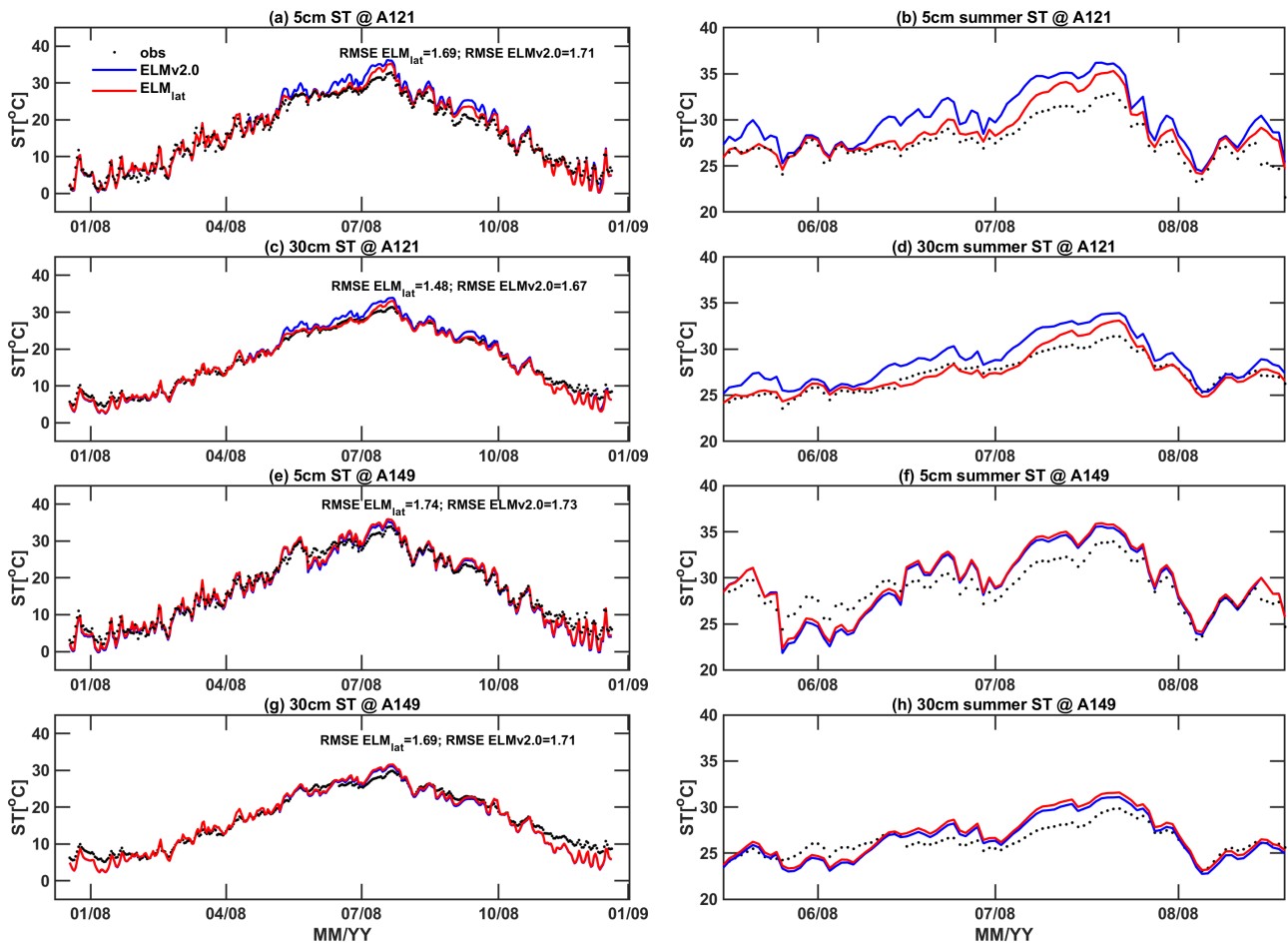

**Figure 12.** comparisons of simulated hourly annual and summer soil temperature at depths of 5 cm and 25 cm for a low land station A121 and a high land station A149: (a) 5 cm annual A121, (b) 5 cm summer A121, (c) 25 cm annual A121, (d) 25 cm summer A121, (e) 5 cm annual A149, (f) 5 cm summer A149, (g) 25 cm annual A149, (h) 25 cm summer A149, between $ELM_{lat}$ and ELMv2.0

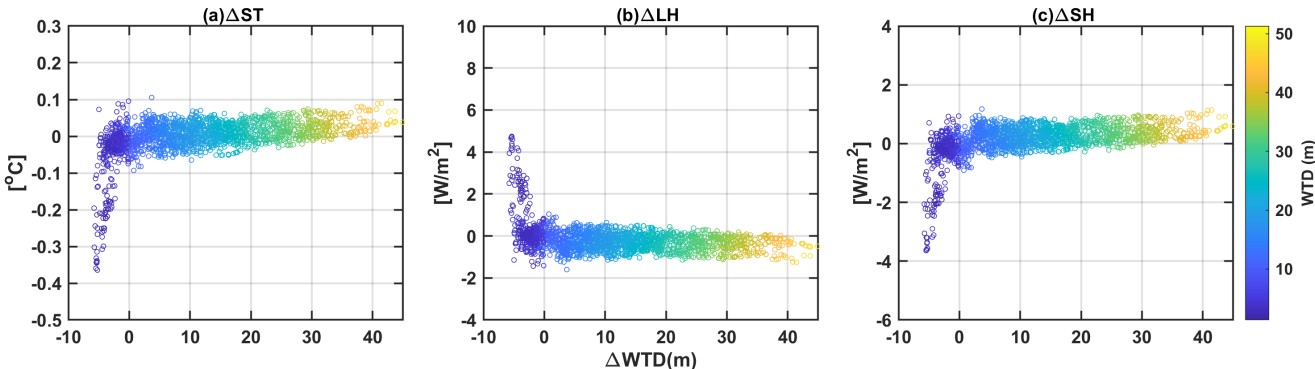

**Figure 13.** (a), (b), (c) are the difference of annual average surface soil temperature, latent heat flux, and sensible heat flux between ELM-lat and ELMv2.0 vs. simulated groundwater depth differences between ELM$_{lat}$ and ELMv2.0; color bar indicates the groundwater table depths simulated by ELM$_{lat}$

the SH differences as discussed previously. The presence of higher soil moisture resulted in cooler summer soil temperature at station A121, whereas lower soil moisture resulted in increased summer soil temperature at station A149. However, both ELM$_{lat}$ and ELMv2.0 overestimated the temperature during July compared with observations, which may be due to the overestimation of incoming solar radiation or the discrepancies of the soil thermal properties between the model and reality at the sampled locations.

Lateral groundwater flow reshapes the groundwater table map and impacts the surface heat fluxes (Figure 13). The most significant changes between ELM$_{lat}$ and ELMv2.0 in heat fluxes occur when ELM$_{lat}$-simulated WTDs are less than 10 [m], where ELM$_{lat}$ simulated generally shallower WTDs than ELMv2.0. As a result, ELM$_{lat}$ simulated lower surface soil temperature and lower SHs, but higher LHs at shallower WTDs, which are consistent with previous discussions. The difference between surface soil temperature, LHs and SHs are nearly linearly correlated with the WTD differences when the WTDs are

less than 10 [m]. At deeper layers, the heat fluxes are not sensitive to changes in WTDs. The maximum differences of simulated surface soil temperature values could reach $-0.3 - -0.4 \ ^\circ C$, the LHs differences could reach $8-10$ [W/m$^2$] and the SHs differences could reach $-4 - -6$ [W/m$^2$], at grids where ELM$_{lat}$ simulated WTDs are less than 5 [m]. The depth at which groundwater can influence the vegetation functioning is dependent on the roots' penetration depths (Fan, 2015). However, since the land use type is quite simple in this watershed (e.g., mainly grass), it is difficult to differentiate the effects of WTD changes on the

vegetation functions for different PFTs.

    For the LWW case, the unsaturated groundwater flux is at a much lower magnitude relative to the saturated lateral flux (Figure 14). The size of the unsaturated flow for LWW is at the magnitude of $\sim 10^{-8}$ [mm/s] while the magnitude of saturated flow is at $\sim 10^{-4}$ [mm/s]. However, since unsaturated lateral flow is closer to the land surface relative to saturated flow, it still plays an important role in regulating the WTD and energy fluxes. Excluding unsaturated flow in ELM$_{lat}$ can result in

the difference of simulated WTD at the range of $\pm 0.04$ [m]. The effect of unsaturated lateral flux on the energy fluxes are

more pronounced right after a rainfall event(Figure A8). Redistribution of soil moisture after the rainfall event significantly influenced the surface energy partitions. However, the redistribution are more confined in local areas which indicates it is not possible for the water to transport from upstream to downstream instantly after the rainfall through unsaturated flow. The energy flux change did not show a similar pattern with the land surface elevations of the whole watershed.

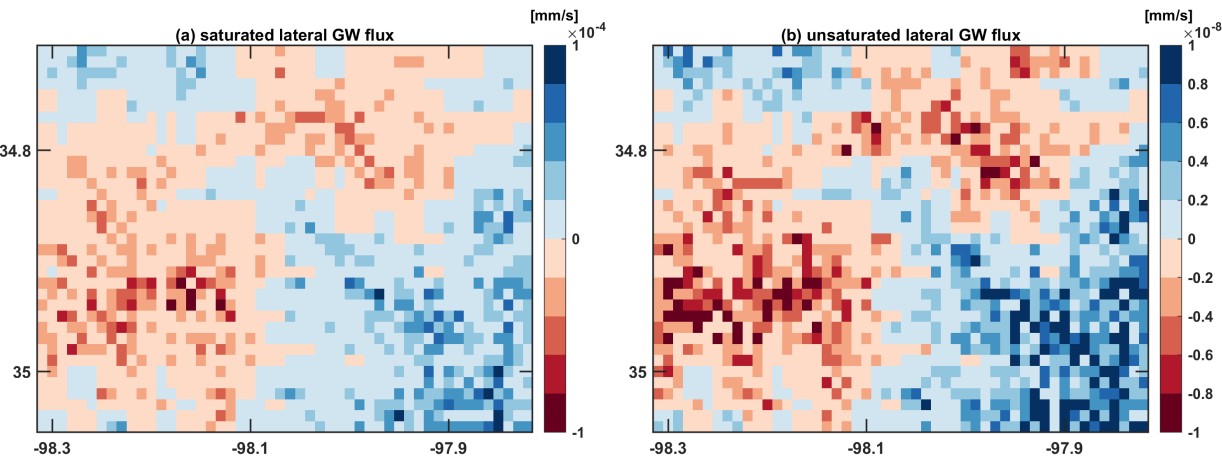

**Figure 14.** The magnitude of (a) saturated lateral flux and (b) unsaturated lateral flux for Little Washita Watershed.

## 3.3 Caveats and future work

A homogeneous anisotropic ratio was assumed in the LWW study given the relatively uniform soil properties. Homogeneous aquifer depth or depth to bedrock (DTB) was also assumed in the LWW study. Brunke et al. (2016) evaluated the effects of using spatially heterogeneous soil thickness and sedimentary deposits (Pelletier et al., 2016) on the simulated hydrological and thermal fluxes in CLM4.5. Incorporating heterogeneous DTB in CLM4.5 has more impacts observed in locations with shallow bedrock than in deep bedrock on the water and energy fluxes, reported by Brunke et al. (2016). Incorporation of variable DTB and evaluation of the heterogeneity of parameters will be performed in the future study.

This study is primarily oriented to validate the new features of the unsaturated and saturated lateral flow in ELM across grids and evaluate the connections between lateral groundwater flow and surface energy fluxes rather than rigidly close the water balance by calibrating all the water balance components against observations. Therefore, water balance and streamflow were not evaluated against observations in this study. A holistic evaluation of the impacts of the lateral groundwater flow on the water and energy balances, as well as the parameter sensitivities, will be tested and evaluated in a much larger domain in future work. In the present work, we have developed $ELM_{lat}$ that runs serially and the model will be extended in the future to use high performance computing to manage the high computational cost of global simulation. Researches found the groundwater temperature can seasonally influence the surface water temperature and soil temperature (Hannah et al., 2004; Qiu et al., 2019).

However, ELM assumes no heat flux boundary condition at the soil bottom, lacks lateral heat diffusion, and does not include advective heat transport. Uncertainties associated with absence of these processes are to be explored in the future.

     Additionally, some anthropogenic activities, e.g., groundwater pumping, irrigation, as well as two way river/groundwater interactions are not incorporated into the current model structure. Extensive groundwater pumping reduced discharge to streams, with 10–23% of watersheds reaching critical environmental flow thresholds, revealed from large scale groundwater modeling

results (de Graaf et al., 2019). We envision a future road map with more holistic representations of hydrological functions building upon the lateral connections of the grid network, not exclusively the lateral groundwater flow, but more realized groundwater and surface water interactions in ELM.

## 4   Conclusions

Regarding the emerging highlights of lateral groundwater flow in the hyper-resolution large scale Earth system modeling, we

developed and validated an inter-grid cell lateral groundwater flow model for both saturated and unsaturated zone in the E3SM Land Model framework.

     By incorporating lateral groundwater flow in the ELMv2.0 and modifying the flux terms based on the non-horizontal terrain, the $ELM_{lat}$ could accurately simulate the soil moisture and WTD dynamics in three idealized hillslope problems validated against PFLOTRAN. During the simulation period, the MAE between the two models is within $1\% \pm 3\%$, and the RMSE is

385 within 0.04. The simulated WTD differences are within $\pm 0.2$ m.

     The developed model was further tested in a realistic watershed, Little Washita Watershed. The simulated WTD by $ELM_{lat}$ showed a strong correlation ($r^2$ = -0.85) with surface topography where higher land has a deeper groundwater table, which agreed well with the WTD pattern in Fan2013 $r^2$ = 0.87 . The effects of lateral groundwater flow on the energy flux partitions were more pronounced at low elevation areas with shallower groundwater tables (i.e., WTD < 10 m). Lateral groundwater

movement from highland area to the valleys cooled down the summer surface soil temperature at low land areas; at high land area, less water slightly increased the summer surface soil temperature. More water in the low land area supported higher LH while reducing the SH compared with the ELMv2.0 simulation. In mountain area with very deep WTDs (> 10 m ), the movement of lateral fluxes have relatively small effects on the surface energy fluxes relative to the effects at the low land area. These results underscore the importance of including lateral groundwater flow in the LSMs, especially in the critical zone to

holistically understand the role of groundwater system on the terrestrial water-energy distribution and its feedback to climate change.

*Code and data availability.*

     Data files and running scripts for model simulations are available at: https://doi.org/10.5281/zenodo.7659300 (Qiu et al., 2023a). $ELM_{lat}$ model are available at:https://doi.org/10.5281/zenodo.7686303 (Qiu et al., 2023b) for the idealized hillslope

problems and at:https://doi.org/10.5281/zenodo.7686381 for the LWW application (Qiu et al., 2023c).E3SMv2.0 (including ELMv2.0) is supported by the Linux system.

The details for running E3SMv2.0 (including ELMv2.0) can be found at:https://e3sm.org/model/running-e3sm/e3sm-quick-start/. $ELM_{lat}$ adopts the same compilation and running approaches. PFLOTRAN can be installed and compiled on Linux, Mac, and Windows systems. The detailed instructions for PFLOTRAN installation and running are provided by the user's guidance
at:https://www.pflotran.org/documentation/user_guide/user_guide.html

## Appendix A: Transmissivity calculation

Following Fan et al. (2007), the transmissivity is calculated for two different cases: case 1: the water table is above 1.5 [m] depth and case 2: the water table is below 1.5 [m] depth. The 1.5 [m] depth is chosen since the availability of reliable soil data is only to that depth.
For case 1:

$$T = T_1 + T_2 \tag{A1a}$$

$$T_1 = \Sigma K_m \Delta z_m \tag{A1b}$$

$$T_2 = \int_0^\infty K dz' = \int_0^\infty K_0 \exp(-z'/f) dz' = k_0 f \tag{A1c}$$

where $m$ is soil layers number between the water table and the 1.5 [m] depth, $K_m$ [m/s] denotes the hydraulic conductivity at
415 layer m, $f$ [m] is the efolding depth which the calculation will be explained later.

For case 2:

$$T_2 = \int_d^\infty K_0 \exp(-z'/f) dz' = k_0 f \exp \left( - \left[ \frac{z - h - 1.5}{f} \right] \right) \tag{A2}$$

where $z$ [m] is the land surface elevation, $h$ [m] is the water head, $d(= z - h - 1.5)$ [m] is the distance between the water table depth and the 1.5 m depth. The calculation of the efolding depth $f$ is based on an empirical equation:

$f = \dfrac{a}{1 + b\beta} \tag{A3}$

where $a$ and $b$ are constants, and $\beta$ is the terrain slope. In this study, $a = 120$ [m] and $b = 150$ [m] were used following the best fits estimated by  Fan et al. (2007),

## Appendix B: Soil parameter transformation

ELM uses the Clapp-Hornberger formula (Clapp and Hornberger, 1978) for paramerizing the soil water retention propertiies,
as shown in equation (10), while PFLOTRAN uses the Brooks Corey (Brooks, 1965) formula to parameterize the soil water

retention properties. For the Burdine-Brooks-Corey water retention formula, the effective saturation, $s_e$ [-], can be expressed as:

$$s_e = \frac{s_w - s_r}{1 - s_r} \tag{B1a}$$

$$s_e = \begin{cases} \left(\frac{-P_c}{P_c^0}\right)^{-\lambda}, & if\, P_c < P_c^0 \\ 1, & if\, P_c \geq P_c^0 \end{cases} \tag{B1b}$$

where $s_w$ [-] is the soil saturation and $s_r$ [-] is the residual saturation; $P_c$ [Pa] is the capillary pressure, $P_c^0 = 101325$ [Pa] is the air entry pressure, and $\lambda$ [-] is a parameter. In unsaturated conditions and assuming $s_r = 0$:

$$p_c = p_c^o s_w^{-1/\lambda} \tag{B2}$$

PFLOTRAN uses the relative permeability, $k_r$, instead of hydraulic conductivity involved in the calculation of the Richards equation, which the relative permeability can be calculated as:

$\quad k_r = (s_e)^{3+2/\lambda} \tag{B3}$

where $\lambda$ is a parameter.

The transformation of the parameter between the two models can be expressed as

$$\lambda = \frac{1}{B} \tag{B4}$$

By performing this transformation and assuming $s_r = 0$, the two water retention curves are very similar except for the differ-
440 ence of using terms of hydraulic conductivity against effective permeability.

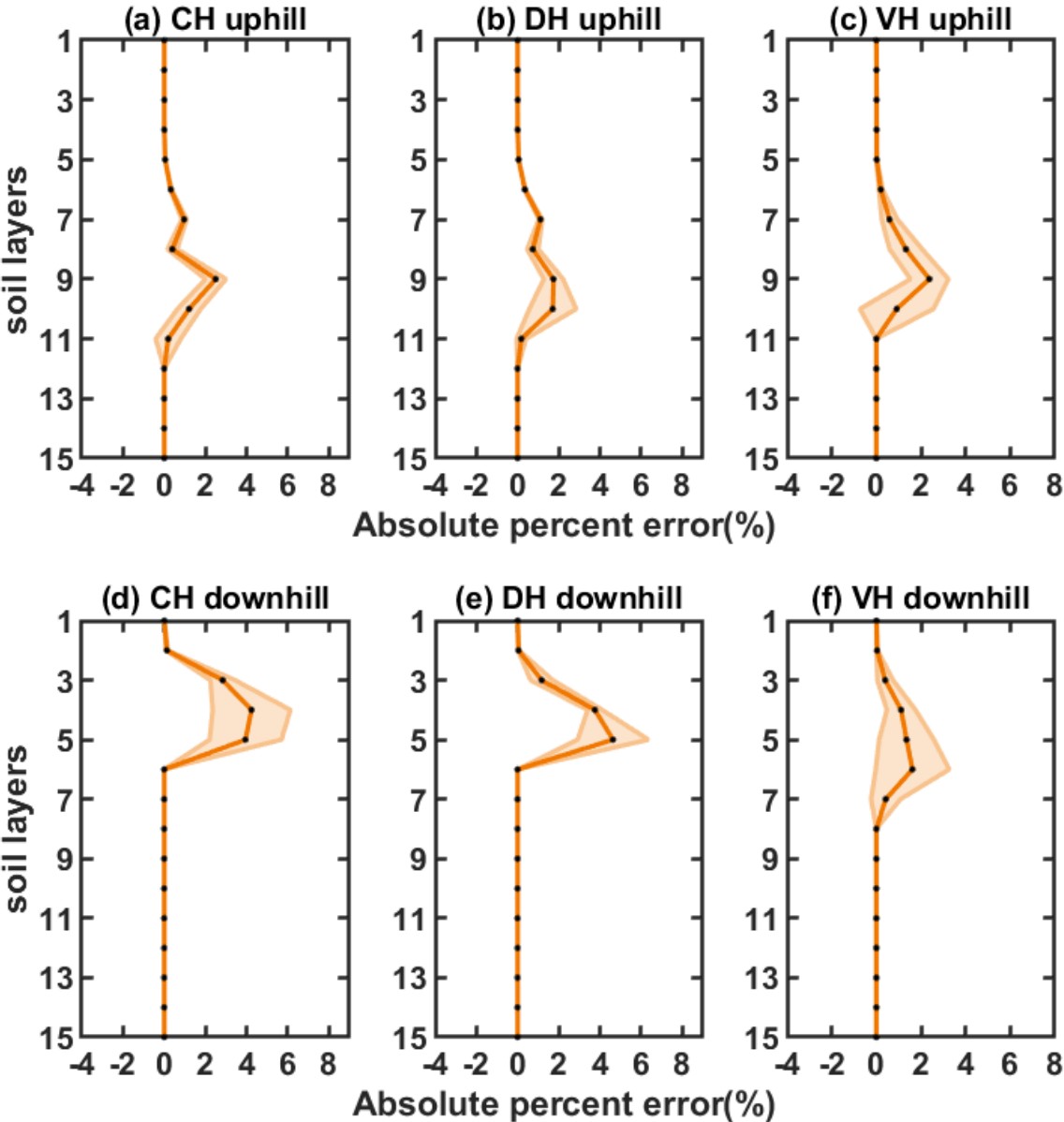

**Figure A1.** Vertical absolute percent errors between ELM$_{lat}$ and PFLOTRAN at the most uphill column for (a) CH, (b) DH, (c) VH and the most downhill column for (d) CH, (e) DH, (f) VH, for each layer at the $80th$ simulation day

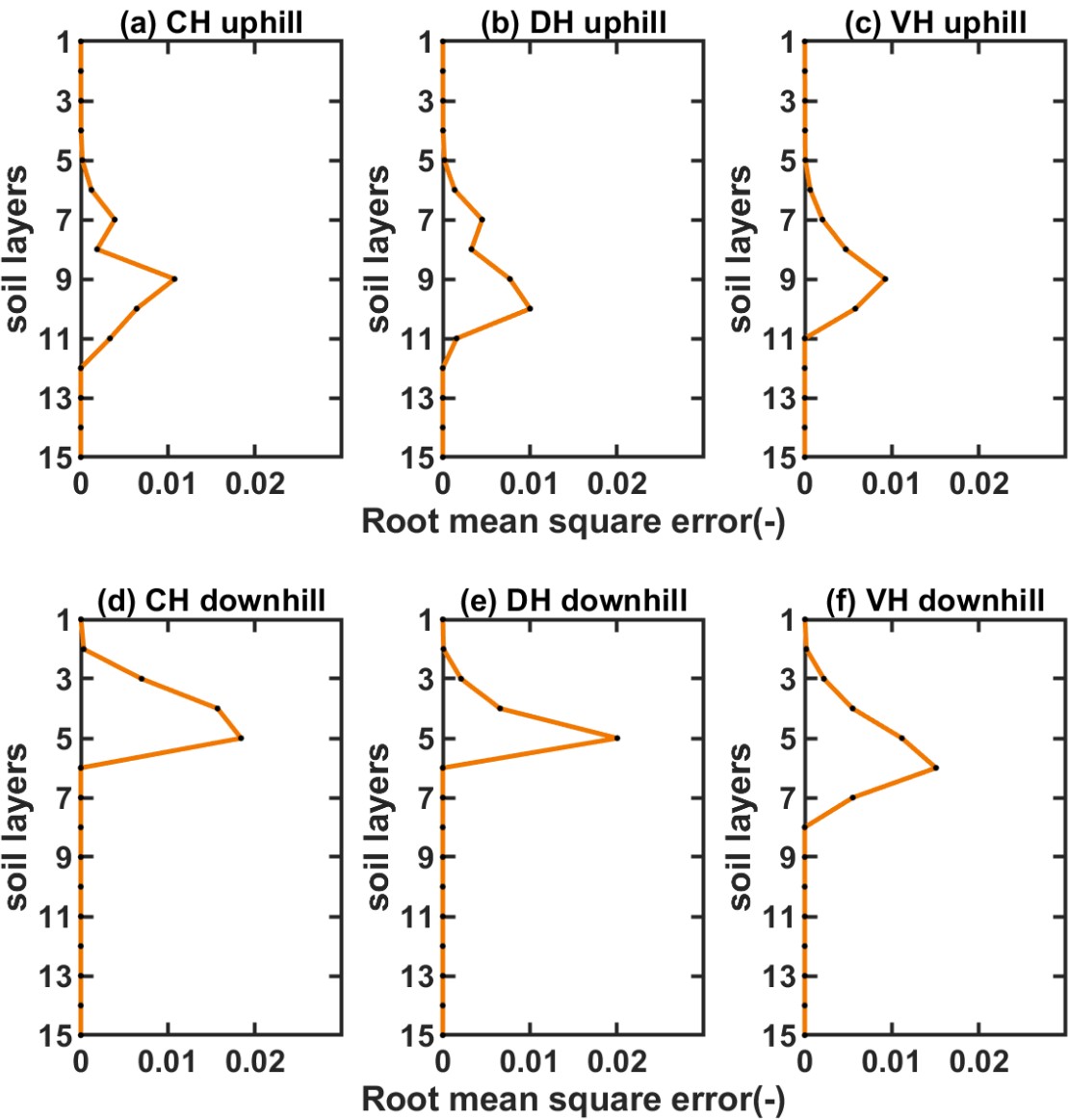

**Figure A2.** Vertical root mean square errors between $ELM_{lat}$ and PFLOTRAN at the most uphill column for (a) CH, (b) DH, (c) VH and the most downhill column for (d) CH, (e) DH, (f) VH, for each layer at the $80th$ simulation day

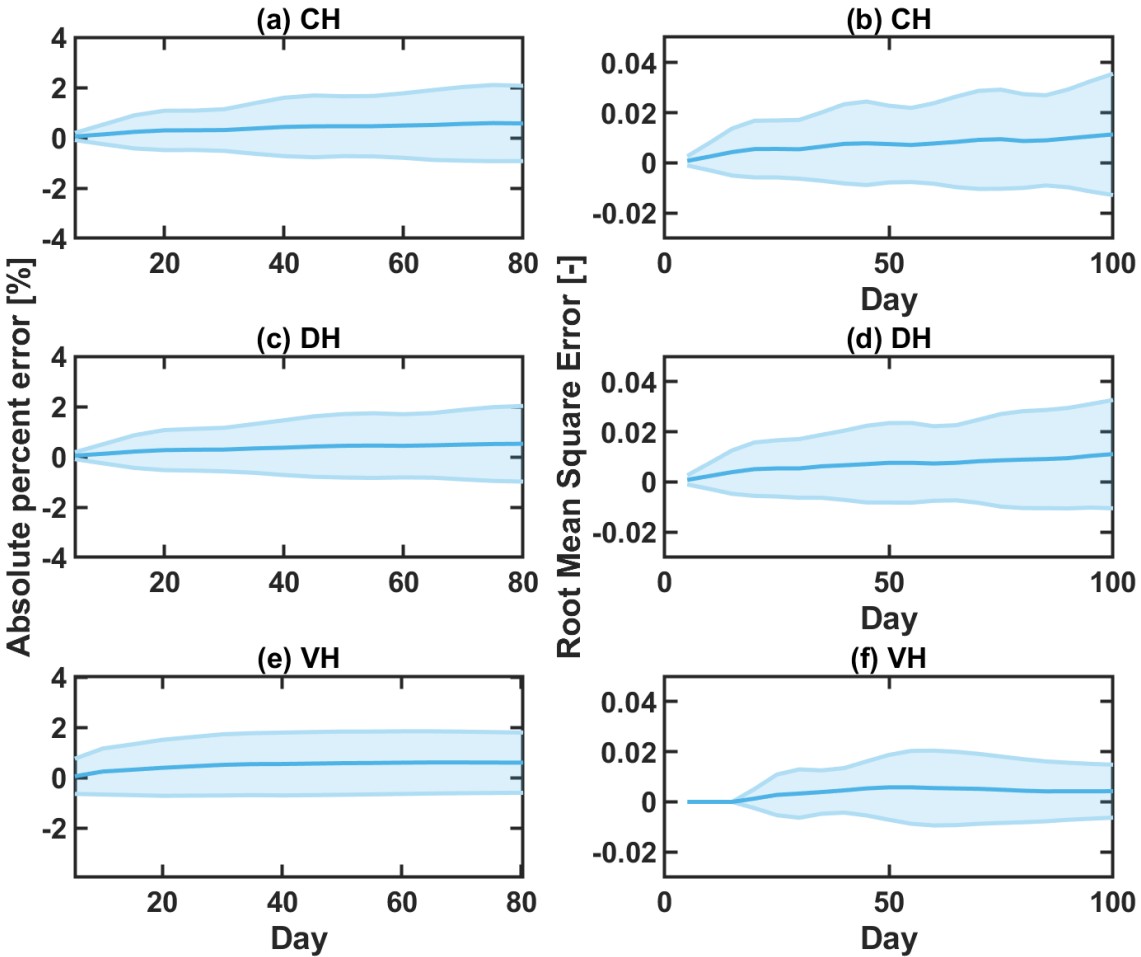

**Figure A3.** The mean absolute error (a,c,e) and root mean square error (b,d,f) of simulated soil moisture between $ELM_{lat}$ and PFLOTRAN for the top ten soil layers during the simulation period

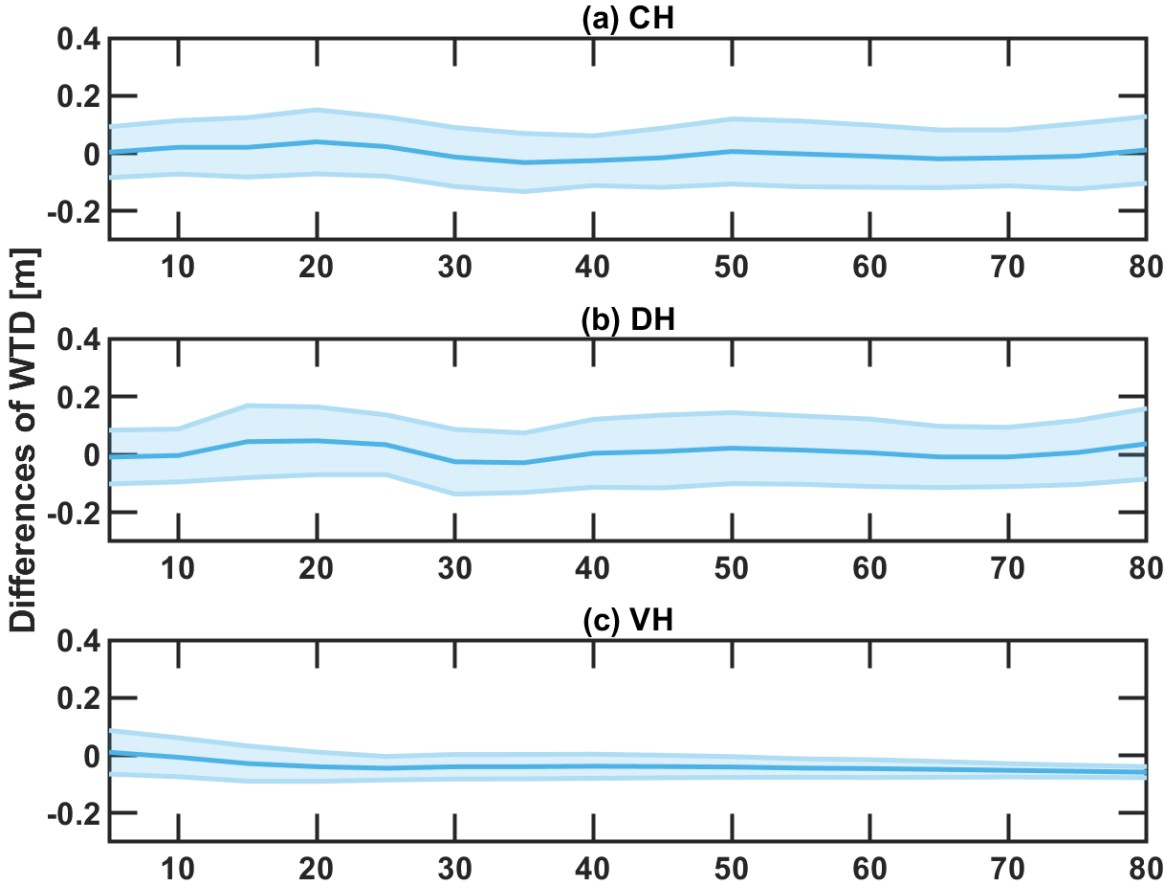

**Figure A4.** The differences of simulated groundwater table depths between ELM$_{lat}$ and PFLOTRAN during the simulation period

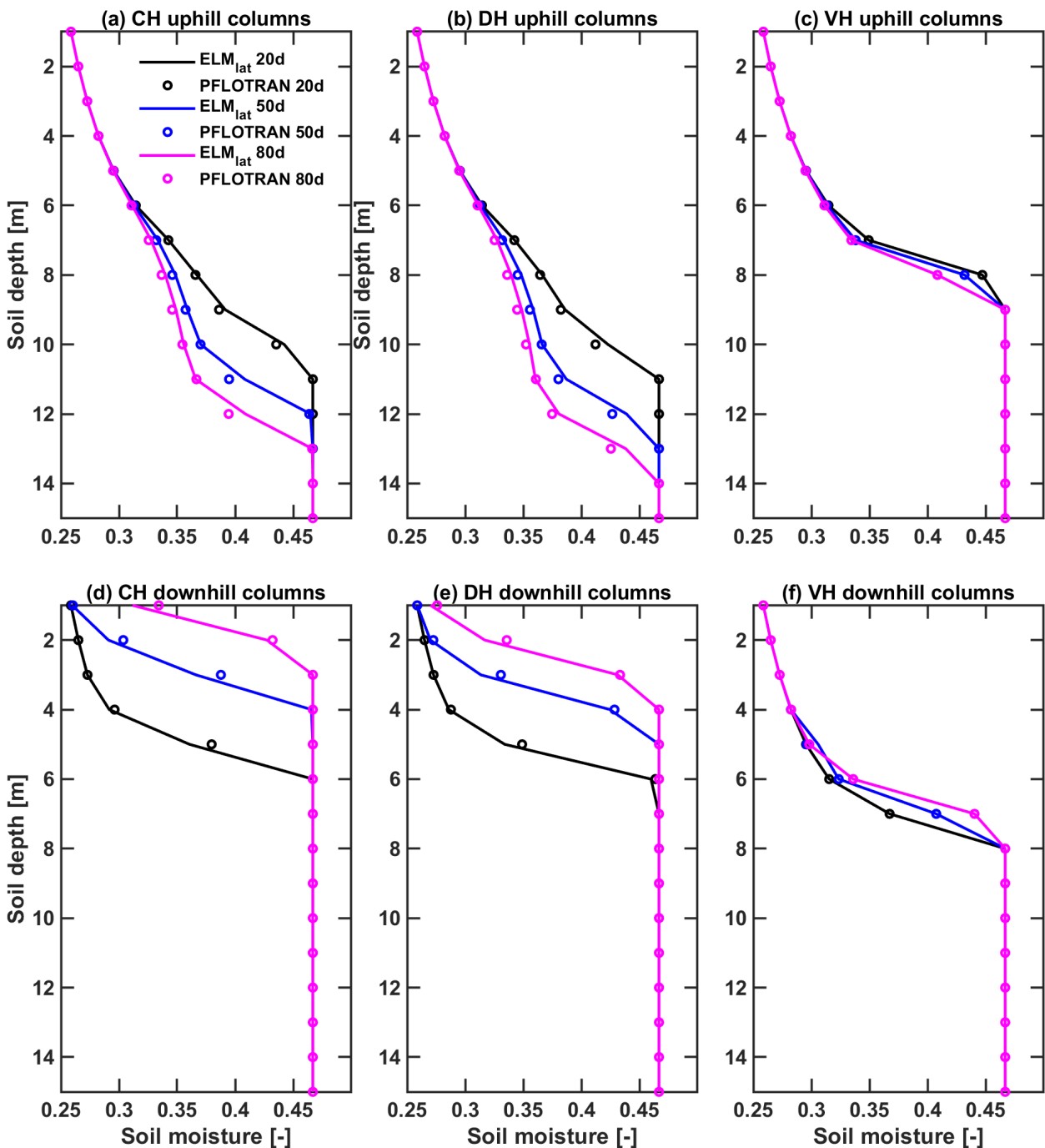

**Figure A5.** ELM$_{lat}$ vs. PFLOTRAN simulated mean soil moisture profile for the ten uphill columns of (a) convergent hillslope (b) divergent hillslope, (c) V-shape hillslope, and ten downhill columns of (d) convergent hillslope, (e) divergent hillslope, (f) V-shape hillslope, at 20th, 50th and 80th simulation day, respectively. For convergent hillslope and divergent hillslope, the anistropic ratio is 10; for V-shape hillslope, the anistropic ratio is 1.

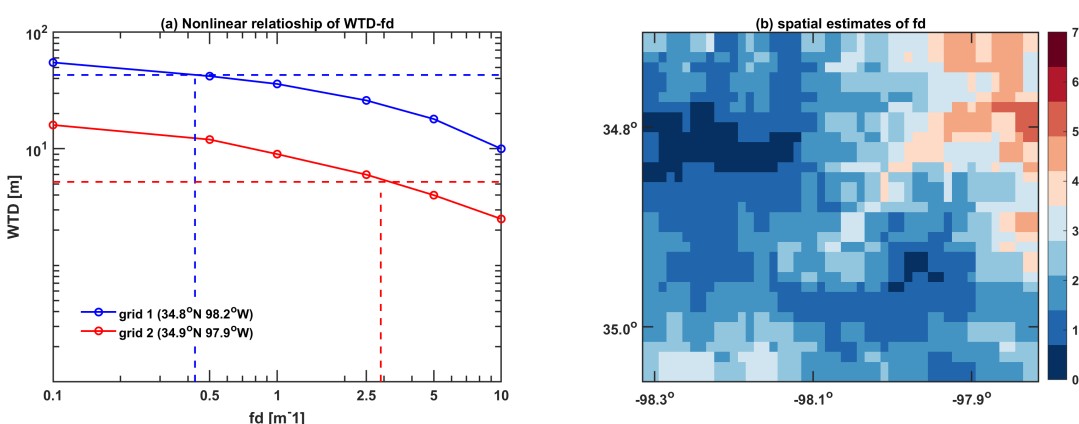

**Figure A6.** (a) An example of establishing the nonlinear functional relationship of WTD-fd and (b) the spatial distribution of calibrated fd values

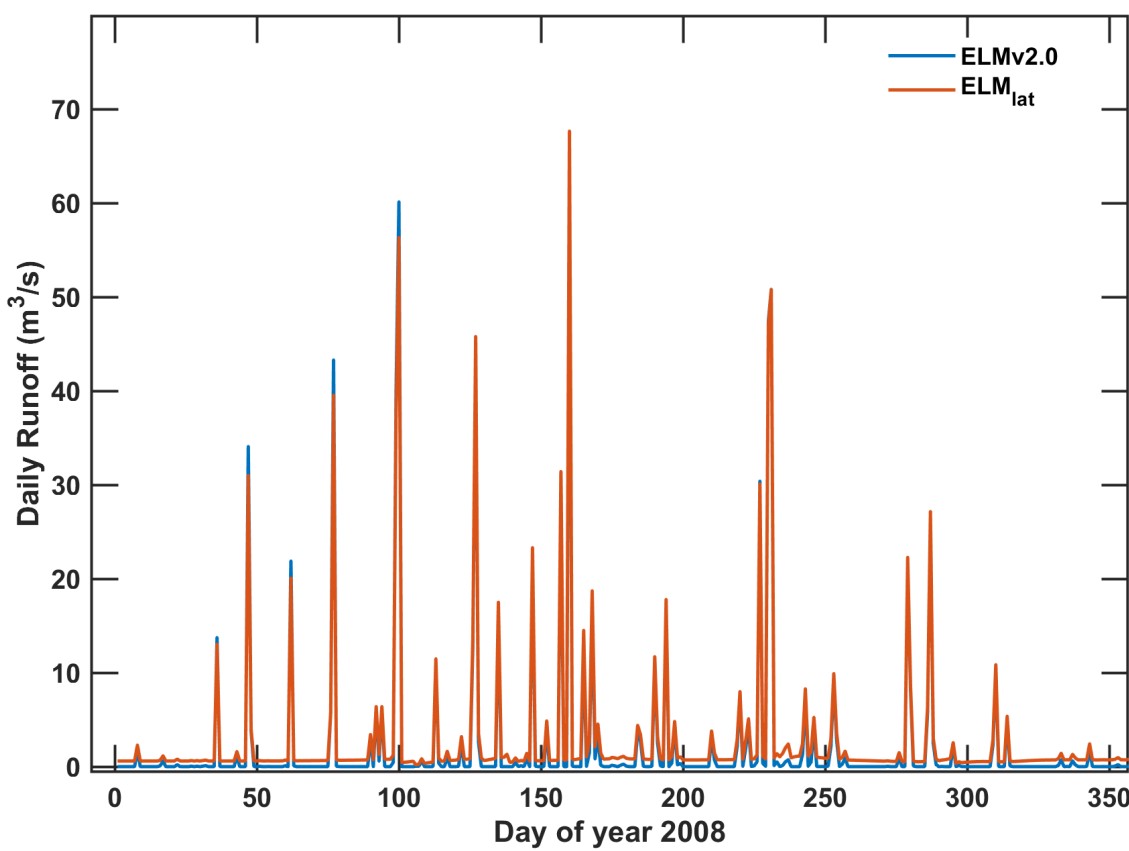

**Figure A7.** Comparison of simulated runoff between ELM$_{lat}$ and ELMv2.0.

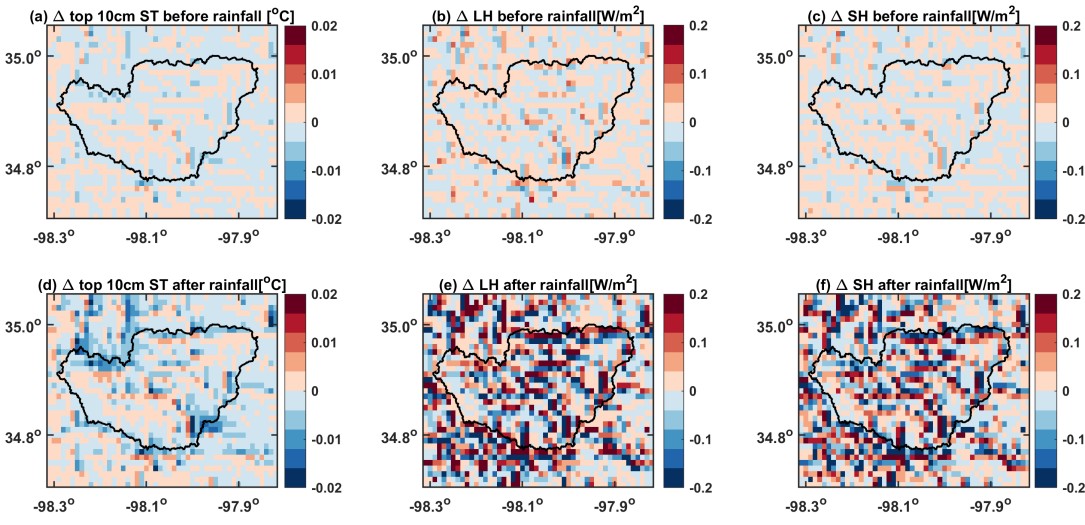

**Figure A8.** The energy flux changes for the top 10 cm soil of Little Washita Watershed after excluding unsaturated later flux for the day before and after a rainfall event during July 6-July 7: (a), (b) and (c) are the soil temperature, latent heat flux and sensible heat flux changes the day before the rainfall event; (d),(e),(f) are the soil temperature, latent heat flux and sensible heat flux changes the day after the rainfall event.

*Author contributions.* HQ designed the study, developed the model, processed the data, and prepared the original manuscript. GB designed the study, discussed the results, and edited the paper LL, DH, and DX processed the data, contributed to subsequent analysis, and helped write and edit the manuscript.

*Competing interests.* The contact author has declared that none of the authors has any competing interests.

*Acknowledgements.* The reported research was conducted at Pacific Northwest National Laboratory, which is operated for the U.S. Department of Energy by Battelle Memorial Institute under contract DE-AC05-76RL01830. The simulations for this research were performed on the CompyMcNodeFace, a Department of Energy Office of Biological and Environmental Research's computer system

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
