# Peer review of "Development of Inter-Grid Cell Lateral Unsaturated and Saturated Flow Model in the E3SM Land Model (v2.0)"

_EGUsphere, 2023_

## Author Comment (AC1)

RC1:

This is a review of "Development of Inter-Grid Cell Lateral Unsaturated and Saturated Flow Model in the E3SM Land Model (v2.0)" by Qiu et al.

The authors describe the development of a saturated lateral flow parameterization for between-gridcell water movement. The modified model is compared to a fully 3d subsurface flow model.

Derivation of equations

In general, I thought that the derivation of the moisture movement equations could be improved. Around line 105, the authors reference Oleson et al. [2013], which describes the conservation of mass in one dimension that is used in the CLM model. But their equation 7 is not what is found in Oleson et al., and instead is written in the general 3d form. The authors here use the index n to denote different control volumes, before switching to an index k used to describe the layers of the 1d soil structure. I think it would be more clear to either 1) describe the 3d equations first, then show how the specific 1d case leads to the ELM/CLM equations described in Oleson et al., or follow Oleson et al. and then show how the 3d equations are used to define the new term in equation 14.

Response: Thank you for the suggestions. In the revised manuscript, we updated Section 2.1 and have adopted the suggested option-1. Specifically, we described the general 3-D soil water movement equations first and then introduced the simplified 1-D equation used by ELM/CLM. Please see Equations 1-14 in the revised manuscript. Additionally, we have updated the figure 1 in the revised manuscript to explain the implementation details of the computation of lateral fluxes in the unsaturated and saturated zone.

Regarding the equations describing the lateral flux, e.g. 14 and following, the areas used to convert between fluxes and volumes should be clarified to show whether they are actual surface areas, or projected areas. The appropriate area is the area that is normal to the direction of the flux. This is why I found the description of equation 16 confusing. I don't think describing the fluxes relative to z' (the rotated z axis) make sense. This is still a 1d column model, with the nodes aligned vertically, therefore the fluxes in the column are all in the vertical. Presumably the coupling to the atmosphere also occurs in the vertical. It would be more clear to me to note that a cosine arises in equation 16 due to the projected area of the surface being smaller than the surface area by a cosine factor.

Response: This is a very helpful suggestion. The volumetric flux is the dot product of the flux vector and the area vector which either the flux can be adjusted by the cosine of the angle or using the area that is normal to the direction of the flux. Given the fluxes in the column are all vertical, we applied the cosine to the area, indicating the projected area of the surface. Please see equation (16) in the revised manuscript.

I would also like to know if the presence of z in equation 15 is correct. I understood the gravity term in the modified Darcy equation to be the sin(theta) term.

Response: The presence of z is for reference of elevation. This equation is also used by ParFlow, please see Equation (7) in Maxwell (2013). This equation is originally derived from Childs (1971). We have added this reference in the revised manuscript.

Why does equation 17 not have a similar form as equation 15? Also, I would like to see the calculation of the transmissivity T described here rather than simply referenced.

Response: The reviewer is correct that equation 17 for the lateral saturated flux should follow a similar form as equation 15 for the lateral unsaturated flux. We corrected this mistake in the revised manuscript, as well as in the model code. All simulations were rerun using the corrected model. We found this correction improved the model results while benchmarking against the PFLOTRAN for the three idealized hillslope problems. We also added details of the transmissivity calculation in Appendix A.

What is the size of the contribution of the unsaturated lateral flow term, for both the benchmarking and application simulations? It would be useful to indicate the relative importance of this term, and whether this impacts the simulations significantly. It would seem to be straightforward to turn off this flux to test this issue.

Response: Relative to unsaturated flow, the saturated flow plays a more dominant role in magnitude. We have added the text in the revised manuscript related to the results of the magnitude of the unsaturated lateral flow and saturated lateral flow. Please see Figure 8 for the idealized problems and Figure 14 for the little Washita case. Additionally, we evaluated the change of energy flux results after turning off the unsaturated lateral flux through a rainfall event, please see Figure A8.

Evaluation over lww

A comparison to a LWW PFLOTRAN simulation would have been interesting. Given that PFLOTRAN was used in the benchmarking section, why was it not used in the evaluation section?

Response: While it is tempting to investigate the tradeoffs between the computational cost and simulation accuracy by comparing ELM-PFLOTRAN (i.e.,3D variably saturated subsurface model)and ELM-Lat (i.e., a simplified approach to include lateral unsaturated and saturated flow), such a comparison is beyond the scope of this work. In the LWW application, it can be extremely computationally expensive to drive PFLOTRAN with the prescribed hydrologic fluxes computed by ELM under extremely dry conditions. For example, the ELM-simulated evapotranspiration, which is transient and varies vertically in the soil, could be large for a single drier control volume within the 3D domain, thus resulting in PFLOTRAN to take extremely small timesteps. Therefore a comparison for LWW between the two models is beyond the scope of this study.

The WTD map in figure 8 shows that the addition of lateral flow helps to better resolve the uphill/downhill differences in water table depth, but there are still significant differences relative to the Fan WTD map. The authors note that calibration of the f_d parameter to give a better match to the Fan WTD map may not be fruitful due to differences in climate forcing. But given the relatively large differences, it would be informative to do a sensitivity test for the f_d

parameter. For example, is there a value of f_d that further lowers the water table depth, and better resolves the riparian areas as shown in the Fan map?

Response: In the revised manuscript, we calibrated f_d values for each grid cell instead of using a spatially homogeneous f_d value. According to our sensitivity analysis of f_d, smaller f_d leads to a deeper groundwater level because the subsurface runoff increases. The updated f_d values led to a significant improvement of the WTD simulations when compared to the Fan2013 WTD map. We found the default ELM_v2.0 also showed spatial WTD variations while using heterogeneous f_d values, but the spatial variations are very small to have a clear pattern.

Statements such as "The effects of WTD changes on the energy fluxes were more pronounced at low elevation cells, especially at the stream and its surrounding cells. The delivery of the groundwater through the lateral flow to the valleys supported higher LH while reducing the SH compared with ELMv2.0 which has little spatial WTD variations" do not appear to be well supported by figure 9. Instead of highlighting the differences between the uphill and downhill areas apparent in figure 8, figure 9 shows spatial patterns having broad domain-wide patterns. Why do the water table patterns in figure 8 show much more structure? For example, larger LH values do not appear consistently in the riparian areas. Similarly, the patterns in the difference maps only show scattered points rather than a clear riparian pattern. Why is this?

Response: The water table patterns shown in figure 8 are primarily driven by the terrain. The hotspots shown in the difference maps are mostly located at low elevation areas, where the influence of the WTD changes is more prominent. That's the reason we made the statement "The effects of WTD changes on the energy fluxes were more pronounced at low elevation cells, especially at the stream and its surrounding cells. The delivery of the groundwater through the lateral flow to the valleys supported higher LH while reducing the SH compared with ELMv2.0 which has little spatial WTD variations." However, the energy fluxes can be determined by many different factors, including the land use type, soil properties and local climate forcings, etc.. Some of them can be dominant processes which the effect of water table change on the energy fluxes can be relatively small. That could explain why the patterns in the difference maps only show scattered points rather than a clear riparian pattern.

The comparisons to observations (figures 10 and 11) do not seem to add much insight into the relative model behaviors. Given the authors choice to not calibrate the models, I don't think it can be stated that the differences between the observations are due to model structure. For example, the A121 differences in figure 10 might be smaller if the f_d parameter in ELMv2.0 model had been calibrated. The statement that both models "were able to capture the major fluctuations and wetting/drying cycles of soil moisture (SM)" seems over-stated. The rain events are generally captured, but the magnitude of the response, and the dry-down rate is generally poor. What information are the authors trying to give to the reader with these figures? Similarly for figure 12; one does not need to perform a model simulation to be aware that shallower water tables will typically have colder temperatures, higher LH, and lower SH than deeper water tables. Any two model versions having different water table depths would presumably show this behavior. I don't see that this figure adds any additional insight to the results.

Response: In the revised manuscript, we calibrated the f_d value to match the Fan2013 water table map and get improved results, please see Figure 9 and shown as below:

[Figure]

. And we found we plotted the observed soil moisture using the wrong station data in Figure 10(b), we corrected this in the revised manuscript. However, the simulated dry-down rates of soil moisture results still do not show perfect performance in comparison with observations. It should be noted that the simulated results represent the model behavior of the whole 1-km grid, while the measurements are taken at a single point. It is probable that the soil parameters for the 1-km grid couldn't represent the soil property of the single point. Therefore, we add this statement 'However, the dry-down rates of soil moisture results are not perfectly captured by both ELM$_{lat}$ and ELMv2.0.' and add the explanations. Please see line 318-320 .

We agree it may be common sense that shallower water tables will typically have colder temperatures and lower LH in summer, and higher LH than deeper water tables. But as we stated in the introduction section, most state-of-the-art earth system models have not taken lateral groundwater flow into account. It is unknown to what extent the lateral groundwater flow makes a difference and how the difference is manifested in the model results. Especially when the difference is also related to the model resolution. To use LWW as an example, we gain insight at 1 km resolution, the lateral groundwater flow does make a difference in regulating the soil moisture and energy fluxes. However, there are still many associated processes that are not represented. For example, ELM assumes no heat flux boundary condition at the soil bottom, lacks lateral heat diffusion, and does not include advective heat transport.  Uncertainties associated with absence of these processes are to be explored in the future based on the results and our understanding at this stage. Therefore, although we are at a relatively nascent stage in comprehensively understanding the effects of lateral groundwater on the energy/water nexus, those results are still meaningful to underscore the importance of including detailed subsurface processes in new generation earth system models, especially in the context of global change.

---

## Author Comment (AC4)

CC1:

The manuscript is well written, with interesting results. The conclusions can be further strengthened by addressing the following concerns.

(1) The anisotropic ratio for the hydraulic conductivity ($K_x/K_z$) was set as 1 for the CH and DH cases, but it was set as 10 for the VH case, without any justifications. At least, the authors should perform sensitivity tests using the ratio of 10 for the CH and DH cases and using the ratio of 1 for the VH case, and briefly discuss the results.

Similarly, it is unclear what this ratio is for the LWW case and what the justification is (as dx = 1 km here versus 10 m for the three idealized cases).

Response: The average slope of VH case is much lower than CH and DH cases, such that the lateral flow rate of VH case is low given the same hydraulic conductivity. We used a higher anisotropic ratio of the hydraulic conductivity ($K_x/K_z$) for the VH case to accelerate the lateral water movement for the comparisons. In the revised manuscript, we have tested the sensitivity of the model results to the anisotropic ratio with 1 and 10  and added discussions, please see Figure A5 and line 288-295

The anisotropic ratio has a close relationship with soil property because the presence of clay has a strong effect on anisotropy due to its platy mineral form and its low permeability as a unit. The anisotropic ratio for the LWW case is assigned as 10 referred from Fan et al., (2007) based on the primary soil property of LWW. The detailed relationship between anisotropic ratio and soil property can be referred from Table 2 in Fan et al., (2007). We add those details in the revised manuscript. Please see line 242-244

(2) The authors proposed the lateral fluxes for both unsaturated and saturated flow in Eq. (14). To understand the importance of such fluxes, the authors should compare the magnitudes of lateral flux versus vertical fluxes for the unsaturated zone and saturated zone separately for the three idealized cases.

Response: We calculated the lateral fluxes for both unsaturated and saturated flows and presented the results in a new Figure 8 for the benchmarking problems and Figure 14 for the LWW. The saturated lateral flow plays a more dominant role than unsaturated lateral flow.  Hydraulic conductivity is nonlinearly dependent on soil saturation conditions and varies significantly with soil properties. The scale difference of the hydraulic conductivity between unsaturated flow and saturated flow is the primary reason for the magnitude difference between unsaturated and saturated lateral flux.

(3) For the TWW case, while it is probably acceptable not to compare the results with observed streamflow, the authors should at least compare runoff time series between the two simulations. For instance, how does the lateral flux affect the timing of peak runoff?

Response: We added the comparisons of runoff time series in Figure A7. Including the lateral flux decreased the peak runoff and increased the lower runoff. In addition, the timing of peak runoff is not changed in ELM_lat.

Minor comments:

(4) Line 189: porosity is 0.43 but soil moisture is greater than 0.43 in Fig. 5. Clarify.

Response: The porosity was incorrectly reported in the original manuscript and the correct porosity should be 0.467. We have fixed this error in the revised manuscript. Please see line 189

(5) Line 219: explain how you obtain the atmospheric forcing data at 1 km grid size from the original 1/8 degree data.

Response: The atmospheric forcing was not downscaled from the original 1/8 degree to 1 km, so the resolution of the forcing data is coarser than the surface data. We have clarified this in the revised manuscript at line 223.

(6) Line 225: revise "Google Earth Engine (Gorelick et al. 2017)" by "Google Earth Engine (GEE; Gorelick et al. 2017)"

Response: We have corrected it in the revised manuscript at line 230.

(7) Provide and briefly discuss the correlation between Fig. 8b and Fig. 8c. Also provide simple statistics (e.g., root mean square errors) in each panel in Figs. 10 and 11.

Response: we provided the correlation between Fig.8b and Fig.8c and added discussions, please see line 305-308

And we added the RMSE (root mean square errors) in each panel in Figs 10 and 11(Now Figs 11 and 12).

References:

Childs, E.: Drainage of groundwater resting on a sloping bed, Water Resources Research, 7, 1256–1263, 1971.

Henderson, F. M., and R. A. Wooding. "Overland Flow and Groundwater Flow from a Steady Rainfall of Finite Duration." Journal of Geophysical Research (1896-1977) 69, no. 8 (1964): 1531–40. https://doi.org/10.1029/JZ069i008p01531.

Balsamo, G., Beljaars, A., Scipal, K., Viterbo, P., Hurk, B. van den, Hirschi, M., and Betts, A. K.: A Revised Hydrology for the ECMWF Model: Verification from Field Site to Terrestrial Water Storage and Impact in the Integrated Forecast System, Journal of Hydrometeorology, 10, 623–643, https://doi.org/10.1175/2008JHM1068.1, 2009.

Maxwell, R. M.: A terrain-following grid transform and preconditioner for parallel, large-scale, integrated hydrologic modeling, Advances in Water Resources, 53, 109–117, https://doi.org/10.1016/j.advwatres.2012.10.001, 2013.

Fan, Y., Miguez-Macho, G., Weaver, C. P., Walko, R., and Robock, A.: Incorporating water table dynamics in climate modeling: 1. Water table observations and equilibrium water table simulations, Journal of Geophysical Research: Atmospheres, 112, https://doi.org/10.1029/2006JD008111, 2007.

---

## Author Comment (AC5)

RC2:

The authors have the original idea of considering horizontal groundwater flow in the source/sink terms of the equation. While these are interesting results, I believe the paper could be improved by considering the following points.

About the equation

To clarify the difference between ELMlat and PFLOTRAN, it should be noted that the unsaturated horizontal flow is the results of one previous step.
Response: the lateral flux is based on pressure/theta values from the previous time step, we are using an explicit time integration for the lateral flux.

Equation 16 makes sense if flux includes the surface area of the terrain, but I can't determine that with the current description.
Response: In Section 2.1.1. of the revised manuscript, we have added more descriptions to illustrate this point and modify the equations accordingly. The volumetric flux is the dot product of the flux and the area, which either the flux can be adjusted by the cosine of the angle or the area that is normal to the direction of the flux should be used. We modified the area by using the projected area. Please see Equation (16) in the revised manuscript.

Equation 17 is correct in the equation itself, but does it not have to account for slope gradient as in equation 15? Need a description of why unsaturated horizontal flow is considered but saturation is not.

Response: The first reviewer also raised the same question, and we corrected the equation in the revised manuscript, as well as updated the code. The slope gradient is also considered in the saturated lateral flow calculation, please see Equation (17) in the revised manuscript. All simulations were rerun using the updated code. We found implementing the corrected Equation 17 improved the model performance of $ELM_{lat}$ against PFLOTRAN for the three idealized benchmark problems, as shown in Figure 5.

Regarding the model benchmark

The authors mentioned that the moisture retention curves used in ELM and PFLOTRAN are different, but why not show how much they differ? Showing the results of the fitting would be helpful for the discussion.

Response: ELM uses the Clapp-Hornberger formula to parameterize the soil water retention curves while PFLOTRAN uses the Brooks-Corey formula. We have added the descriptions of the two formulas and how the parameters are transferred from one to the other in Appendix B. By assuming the residual soil water equals zero, the two formulas are actually very similar. In the revised manuscript, We don't attribute the difference between $ELM_{lat}$ and PFLOTRAN to the differences of retention curves.

Regarding the results

What is the reason for using top 5 layers in the comparison with PFLOTRAN as far as Figure 5 is concerned, I think it would be a fair assessment to include up to about top 10 layers. Also, what is the reason for using MAE? It tend to be small because the denominator is larger than the numerator. Why not present other indicators as well?

Response: The reason for choosing the top 5 layers for the comparison is because for some soils columns the layers 6-10 are saturated for both $ELM_{lat}$ and PFLOTRAN during the simulation period. We now compare the top 10 years instead in the revised manuscript. Please see the updated Figure 5. Meanwhile, we have added the Root Mean Square Error as an evaluation metric in Figure A2.

I understand that Figures 6 and 7 are ELMlat-PFLOTRAN. If so, some discussion of the CH and DH soil moisture and groundwater table results is needed; ELMlat has less soil moisture on the down hill and more on the up hill than PFLOTRAN, even though the groundwater table is too flat compared to PFLOTRAN. Is this solely due to the water retention curve? Need to describe the difference in equations and the effect of the solution method.

Response: We added more details about the water retention curve formulas used in the two models. $ELM_{lat}$ has less soil moisture on the downhill and more on the uphill than PFLOTRAN, and the groundwater table is actually steeper compared to PFLOTRAN which can be seen from Figure 7 (d-f). The Water Table Depth (WTD) is the distance between the groundwater table elevation and land surface elevation. $ELM_{lat}$ has less soil moisture on the downhill and higher WTD values (lower groundwater table), but more soil moisture on the uphill and lower WTD values (higher groundwater table), which are consistent with previous discussions. Moreover, we also tested the sensitivities of the model results in response to different anisotropic ratios, as shown in Figure A5.

Regarding Figure 9, from Figure 8, the groundwater table is deeper near the watershed boundary, but even in such locations, does the soil temperature decrease, LH increase, and SH decrease? Are these results consistent? Results and discussion on whether horizontal unsaturated flow has an effect and how the seasonal planar distribution varies are needed.

Response: We made a mistake while adjusting the color map in Figure 9. In the revised manuscript, we replotted the figure with updated results using calibrated f_d values. At locations near the watershed boundary, the groundwater table is deep such that the effect of groundwater level change, which generally becomes deeper due to lateral flow, on land surface temperature and heat fluxes are relatively small, such that the energy fluxes change are approaching zeros at some of those grids.

The magnitude of unsaturated flow against saturated flow is small. Please see Figure 8 and Figure 14 .

The seasonal effect of lateral groundwater flow on the soil temperature and heat flux is dependent not only on the water movement and groundwater table change but also on the lateral heat flux and the groundwater temperature, for example,. However, ELM assumes no heat flux

boundary at the soil bottom, and we do not have a lateral groundwater thermal transport model at this stage, seasonal analysis of the heat flux change may be biased. Since the main purpose of this study is developing the inter- grid cell lateral flow within ELM, we decided to focus more on the model development and validation. How the seasonal planar distribution varies will be discussed in the future study.

Minor comments

Line 127 may be clearer in Figure 1 than in Figure 2

Response: Thank you for this suggestion. We used the updated Figure 1 to illustrate Line 127 (now Line 128) instead.

I think (d) in Figure 10 is a mistake for A149.
Response: We have corrected this in the revised manuscript.

The equation numbers are not bracketed: lines 106, 116, 126, 128, 143.

Response: We have corrected this in the revised manuscript.

FigureA1 should be the result of all layers, not top 5 layers.

Response: We have modified Figure A1 with results of all the top 10 layers.

FigureA2 does not have results for (d), (e), (f).

Response: We have corrected the caption.

Figure A3 has all figures marked as (a).

Response: We corrected this, shown as Figure A4 in the revised manuscript.

Line 540 is Soil MoistureâBased (I noticed this by accident)

Response: Thank you for pointing this out. We corrected this.

References:

Childs, E.: Drainage of groundwater resting on a sloping bed, Water Resources Research, 7, 1256–1263, 1971.

Henderson, F. M., and R. A. Wooding. "Overland Flow and Groundwater Flow from a Steady Rainfall of Finite Duration." Journal of Geophysical Research (1896-1977) 69, no. 8 (1964): 1531–40. https://doi.org/10.1029/JZ069i008p01531.

Balsamo, G., Beljaars, A., Scipal, K., Viterbo, P., Hurk, B. van den, Hirschi, M., and Betts, A. K.: A Revised Hydrology for the ECMWF Model: Verification from Field Site to Terrestrial Water Storage and Impact in the Integrated Forecast System, Journal of Hydrometeorology, 10, 623–643, https://doi.org/10.1175/2008JHM1068.1, 2009.

Maxwell, R. M.: A terrain-following grid transform and preconditioner for parallel, large-scale, integrated hydrologic modeling, Advances in Water Resources, 53, 109–117, https://doi.org/10.1016/j.advwatres.2012.10.001, 2013.

Fan, Y., Miguez-Macho, G., Weaver, C. P., Walko, R., and Robock, A.: Incorporating water table dynamics in climate modeling: 1. Water table observations and equilibrium water table simulations, Journal of Geophysical Research: Atmospheres, 112, https://doi.org/10.1029/2006JD008111, 2007.

---

## Author Comment (AC6)

RC3:

This manuscript gives a good description of the parameterization scheme of Inter-Grid Cell Lateral Unsaturated and Saturated Flow Model, and evaluates the simulation performance of this scheme. But I think there are still some things that need to be improved to make it better. My detailed comments are as follows:

About the parametric scheme

1.    Line 94. For soil water movement, k is an important parameter. Is the description of equation 3a accurate? I noticed that ELM originated from CESM1.3, and the calculation of k in CESM1.3 is divided into two parts according to soil layer (for details, please refer to Oleson et al., 2013, equation 7.89). Why did this paper modify this? Will such modification have a significant impact on the results?

Response: We did not modify the calculation of vertical hydraulic conductivity. Equation 3a ( Equation 10a in revised manuscript) is a general form of the Clapp-Hornberger equation used by CESM1.3. We just didn't specify the usage of this equation for specific layers.

Line 100. The modification of equation 4 can be confusing. I recommend the author can cite "Yan Yu, Zhenghui Xie, XubinZeng, Impacts of modified Richards equation on climate simulation in the regional climate model RegCM4, Geophys. Res. Atmos., 119, 12,642–12,659, doi:10.1002/2014JD021872, 2014.". It would be more appropriate to explain the reasons for the changes.

Response: Thank you for sharing this reference, which is very helpful. We have included this reference in the revised manuscript. Please see line 160

2.    Line 130. Equation 14, does the direct modification of the tridiagonal equation mean that soil water lateral flow is carried out firstly and then vertical flow is carried out in 3D soil water flow? In fact, however, they happen at the same time. Will this have a significant impact on the results?

Response: The lateral and vertical flows in the unsaturated zone are carried out simultaneously within a single tridiagonal equation. However, it should be noted that the vertical fluxes use an implicit time integration method, while the lateral fluxes are solved using an explicit time integration method. The coupling between the unsaturated and unsaturated is calculated sequentially. This could be one of the factors that contribute to the errors benchmarked against the PFLOTRAN. Nevertheless,results show that the errors are small as shown in Fig. 5 and Fig. A5 Fig with different anisotropic ratios.

3.    Line 137. I am confused about the expression of equation 15. Generally speaking, the soil water lateral flow is driven by soil matrix potential, but the gravitational potential (z) is obviously included in equation 15. Clarify please.

Response: The detailed derivation of equation 15 is described in Childs (1971) and was used by PARFLOW, please see equation (7) of Maxwell (2013). The gravitational potential (z) is for referring the elevation gradient.

4. Line 137. I think the parametric scheme of soil water lateral flow is the focus of this paper, but this part is too simple in this paper. It is important to give a detailed process of the derivation of equations rather than simply referenced. Moreover, it seems that the final parametric equation is not given in equation 14. I suggest: 1) First introduce the soil flow scheme in ELM; 2) Then introduce the soil water lateral flow scheme and your improvement; 3) Finally, the tridiagonal equation including soil water lateral flow is given. Or switch step 1 and step 2.

Response: As suggested by the reviewer, we have reformatted the equation section to describe the general 3-D soil water movement equations first, then introduced the simplified 1-D equation used by ELM/CLM, followed by the new lateral flux term in equation 14. We added more descriptions about the equation derivations.

5. Line 149. The description of equation 17 is simplistic. In fact, in my opinion, equation 17 is just as important as equation 15, so the derivation of 17 should be detailed rather than simply quoted, because I have noticed there are differences between equation 17 and the equation in Fan (2007).

Response: equation 17 is the Darcy's equation based on the extended form of the Dupuit-Forchheimer assumption. This is also cited and used by Parflow, so we decided not to put the detailed derivation in the manuscript since the derivation is long. Instead, we cite the original reference, please see equation (8) of Childs (1971) and equation (17) of Henderson and Woodinq (1964). Moreover, we added the details of transmissivity calculation in Appendix A.

About model benchmarking

In the benchmarking model, I am more interested in the comparison among ELM, ELMlat and PFLOTRAN. This comparison may be important to reveal the importance of soil water lateral flow. and can more directly show the improvement of this paper.

Response: in the benchmarking model, the water table depths in ELM have no spatial changes since homogeneous surface data and forcing is used and there are no connections among the grid cells. Therefore, we only performed comparison between ELMlat and PFLOTRAN

1. Is the lateral flow of unsaturated and saturated soil water equally important? Or is one of them dominant? How much do they contribute to soil water movement?

Response: The saturated lateral flow plays a more dominant role in the benchmarking model simulations. We present the results of the magnitude of the two fluxes in Figure 8 for the benchmarking problems and also for the LWW case in Figure 14. Additionally, we evaluated the change of energy flux results after turning off the unsaturated lateral flux through a rainfall event, please see Figure A8.

About LWW

1.    Line 297. Why did this paper focus on the summer soil temperature?

Response:  The reason for focusing on summer soil temperature is because it is more influenced by the lateral flow. We clarified this in the revised manuscript and also showed annual soil temperature results in the new Figure 12 and zoomed in the summer temperature for better viewing the difference.

2.    For LWW case, authors only verified soil moisture for shallow soils (25cm), but I think the results below 1m are just as important. I think more soil moisture data (such as ERA5-land or satellite inversion data) can be used to verify the simulated soil moisture, soil temperature and other results, rather than just using two stations data. This is important to illustrate the importance of soil water lateral flow.

Response: Thank you for this suggestion. The ERA5-Land dataset, as any other reanalysis dataset, provides estimates which have some degrees of uncertainty. The land-surface model , HTESSEL (Tiled ECMWF Scheme for Surface Exchanges over Land, Balsamo et al., 2009),  used by ERA-5 dataset has no groundwater lateral flow component. The comparison may be biased due to this mismatch. In addition, the resolution of ERA-5 is 9km, the whole domain will only occupy 3x5 grid cells, the impacts of lateral groundwater flow may be small at this scale.Therefore, we did not compare the model results with reanalysis data like ERA5 considering the above reasons. We will consider including intercomparison with other land surface models with lateral groundwater flow included in a much larger domain in the future work.

3.    Figure 8. In addition to the differences between Figure b and Figure c, the resolution of Figure b seems to be lower. Is this related to the modification of the depth of the soil layer? If yes, can more reasonable soil depth distribution be considered in future work?

Response: Thank you for this suggestion. The resolutions of Figure b and Figure c are the same. The modification of the soil layer depth could be one of the reasons resulting in the lower resolution of Figure 8 (b). Heterogeneous and reasonable distributions of soil and aquifer depth will be considered in the future work which are stated in the 'Caveats and future work' section. Other reasons for the low resolution of Figure 8 (b) may include the homogeneous fd value used which is closely related to the subsurface runoff and coarser resolution (1/8 degree) of the NLDAS forcing data than the horizontal model resolution (1 km) resolution. In the revised manuscript, we calibrated the fd values which significantly improved the spatial groundwater table depth simulations against Fan's results, as shown in Figure 9.

Minor comments

1.    Is line 138 and line142 duplicated?

Response: they are different. Line 138 is explaining the angle of slope at the horizontal direction at the land surface cell center, while Line 142 is explaining the angle between the surface of two neighboring cell centers and the horizontal direction. They can be different if the terrain has non-uniform slopes for different cells. We modified the text to make them clearer. Please see line 138 and line 143

2. Line 146. The unit of is m3/s, but the unit of q in line 124 is mm/s. There is an obvious difference between the two. Whether this is a clerical error, if not, please explain.

Response: The saturated flow is volumetric flux with unit $m^3$/s, while the unsaturated flow is rate flux with unit mm/s. We underscored this in the revised manuscript and use Q instead of q for the saturated flux.

3. It will be better to label (a) (b) (c) in Figure 12.

Response: As suggested, we have added the labels in the revised manuscript.